# Abrupt high-latitude climate events and decoupled seasonal trends during the Eemian

J. Sakari Salonen [1], Karin F. Helmens[2], Jo Brendryen[3], Niina Kuosmanen[4], Minna Väliranta [5], Simon Goring [6], Mikko Korpela[1], Malin Kylander [7], Annemarie Philip[8], Anna Plikk[2], Hans Renssen[9,10] & Miska Luoto [1]

The Eemian (the Last Interglacial; ca. 129–116 thousand years ago) presents a testbed for assessing environmental responses and climate feedbacks under warmer-than-present boundary conditions. However, climate syntheses for the Eemian remain hampered by lack of data from the high-latitude land areas, masking the climate response and feedbacks in the Arctic. Here we present a high-resolution (sub-centennial) record of Eemian palaeoclimate from northern Finland, with multi-model reconstructions for July and January air temperature. In contrast with the mid-latitudes of Europe, our data show decoupled seasonal trends with falling July and rising January temperatures over the Eemian, due to orbital and oceanic forcings. This leads to an oceanic Late-Eemian climate, consistent with an earlier hypothesis of glacial inception in Europe. The interglacial is further intersected by two strong cooling and drying events. These abrupt events parallel shifts in marine proxy data, linked to disturbances in the North Atlantic oceanic circulation regime.

[1] Department of Geosciences and Geography, University of Helsinki, PO Box 64, FI-00014 Helsinki, Finland. [2] Department of Physical Geography and the Bolin Centre for Climate Research, Stockholm University, 106 91 Stockholm, Sweden. [3] Department of Earth Science, The Bjerknes Centre for Climate Research, and the Jebsen Centre for Deep Sea Research, University of Bergen, PO Box 7803, N-5020 Bergen, Norway. [4] Department of Forest Ecology, Faculty of Forestry and Wood Sciences, Czech University of Life Sciences Prague, Kamýcká 129, 165 21 Praha 6, Czech Republic. [5] Environmental Change Research Unit (ECRU), Ecosystems and Environment Research Programme, University of Helsinki, PO Box 65, FI-00014 Helsinki, Finland. [6] Department of Geography, University of Wisconsin, Madison 550 M Park St, Madison, WI 53706, USA. [7] Department of Geological Sciences and the Bolin Centre for Climate Research, Stockholm University, 10691 Stockholm, Sweden. [8] Ecosystem and Landscape Dynamics, Institute for Biodiversity and Ecosystem Dynamics, University of Amsterdam, PO Box 94216, 1090 GE Amsterdam, The Netherlands. [9] Department of Natural Science and Environmental Health, University College of Southeast Norway, N3800 Bø i Telemark, Norway. [10] Department of Earth Sciences, VU University Amsterdam, NL-1081HV Amsterdam, The Netherlands. Correspondence and requests for materials should be addressed to J.S.S. (email: sakari.salonen@helsinki.fi)

The Eemian (also Last Interglacial, Marine Isotope Stage (MIS) 5e; ca. 129–116 thousand years (ka) ago) is of special interest as the most recent case of widespread climatic warming in the geological record. Marine sediment records suggest that the Eemian was characterised by an average global annual sea surface temperature of +0.5 ± 0.3 °C above pre-industrial levels[1] and by summer surface warming of up to 4–5 °C above Arctic lands[2]. Also, the Eemian global sea level was an estimated 6–9 m above present, associated with reduced ice sheets in Greenland and possibly in western Antarctica[3]. The Eemian is an imperfect analogue for anthropogenic global warming as the climate forcing was different, with only minor changes in total greenhouse-gas forcing compared to preindustrial[4] but some of the strongest anomalies in orbital forcing in the past 1 million years[5]. However, despite the different forcing, the resultant latitudinal temperature distribution of the Eemian appears to have been similar to projected warming under an optimistic scenario for greenhouse gas emissions[6]. The Eemian thus provides a valuable opportunity to explore the response of the global environment (e.g., biosphere, cryosphere, sea level) to climate warming relative to preindustrial[3,7] and to constrain the role of individual feedbacks[4,8].

Despite decades of work on the Eemian, existing climate syntheses continue to be hampered by fragmented data. This is especially true for land areas where the available temperature proxy data are strongly concentrated in the Northern Hemisphere mid-latitudes (ca. 30–60 °N)[9–12]. In the Northern Hemisphere high latitudes (60–90 °N), the lack of data is due to widespread glacial erosion during the last Ice Age, compounded by poor dating control on the highly fragmented sedimentary record[2,11,13]. Data from the northern high latitudes are especially important, as observations and modelling of modern climate change, as well as data on past warm climate stages, suggest that the Arctic can take a highly distinct trajectory during climatic warming. Typically, the modelled and observed Arctic warming is strongly amplified compared to the Northern Hemisphere average, due to a suite of high-latitude feedbacks involving the cryosphere, vegetation, surface hydrology, and oceanic circulation[2,14]. Moreover, recent studies on proxy datasets from the North Atlantic Ocean and the North-European continent have revealed surprising complexities in the evolution of Eemian climate and environment. These new findings include abrupt shifts in climate and oceanic circulation[7,15–20] and asynchronous hemispheric surface temperature changes[7,8]. However, due to paucity of high-resolution datasets, the geographic expression and causes of these events remain elusive[15,16].

Here we present a high-resolution (sub-centennial) palaeoclimate reconstruction for the Eemian from northern Finland utilising a pollen and plant macrofossil record. At the Sokli site, a local bedrock anomaly has allowed the preservation of a long and continuous sedimentary sequence spanning the last glacial cycle[11,21,22] and the entire Eemian[16], despite the location in the central part of northern European continental glaciations (Fig. 1), providing a benchmark record of Eemian climatic evolution in the high-latitude land areas. The Eemian sequence at Sokli has earlier been the focus of multi-proxy studies (including pollen, diatoms, and chironomids) on a Mid-Eemian cooling event[16], as well as limnology and geochemistry[23,24]. Here we present quantitative reconstructions of July, as well as January mean air temperature ($T_{jul}$, $T_{jan}$) in the Eemian, using an ensemble of pollen–climate calibration models including classical and machine-learning approaches. The chronological constraints are improved by new northern European speleothem datings. Our aim is to resolve the long-term seasonal evolution, as well as possible abrupt changes of Eemian climate in the European high latitudes. Our climate reconstructions provide a robust record of

repeated Eemian abrupt events, and reveal a complex interplay of insolation and marine forcings and ice-sheet dynamics.

## Results

**Fossil record and vegetation changes**. The key fossil proxy data from the Sokli Eemian sequence, including pollen and spores, stomata, non-pollen palynomorphs, and plant macrofossils, are summarised in Fig. 2. The fossil sequence is divided into seven local pollen zones using a multivariate regression tree (Supplementary Fig. 3).

In Zones I (28.41–25.17 m) and II (25.17–24.65 m), pollen percentage values of *Betula* up to 80% (including *B. pubescens/pendula* (downy birch/silver birch; deciduous tree) and *B. nana* (dwarf birch; deciduous shrub)) indicate the presence of pioneering birch vegetation. Abundant pollen of *Juniperus* and the presence of a variety of light-demanding herbs (*Artemisia*, Chenopodiaceae, *Polygonum bistorta* type, and *Saxifraga oppositifolia* type) suggest the forest remained relatively open. The sandy/silty sediments of Zone I and early Zone II are interpreted to represent a deglacial ice-lake stage. Zone III (24.65–23.40 m) saw dynamic forest development, from open pine forest (*Pinus* pollen up to 75%) to mixed boreal forest with spruce (*Picea abies*) and then back to relatively open larch (*Larix* sp.) and pine dominated forest.

Zone IV (23.40–22.51 m) corresponds with the Tunturi cooling event defined in ref.[16]. In the pollen record, the event is manifested by a sudden increase of *B. pubsescens/pendula* pollen from 20 to 50%, and increased *Juniperus* and lycopods, indicating the replacement of boreal forest by sub-arctic, open birch–pine forest.

The onset of Zone V (22.51–17.98 m) represents the abrupt return of mixed boreal forest with spruce and larch. *Corylus* becomes near continuously recorded, with relative pollen abundances (ca. 1%) only seen in Finnish surface-lake sediments samples within the modern occurrence limits of hazel[25], suggesting local presence. The sparse pollen of *Quercus* and *Ulmus* were most probably long-distance transported. *Pinus* pollen reaches maximum values (ca. 75%) about halfway through

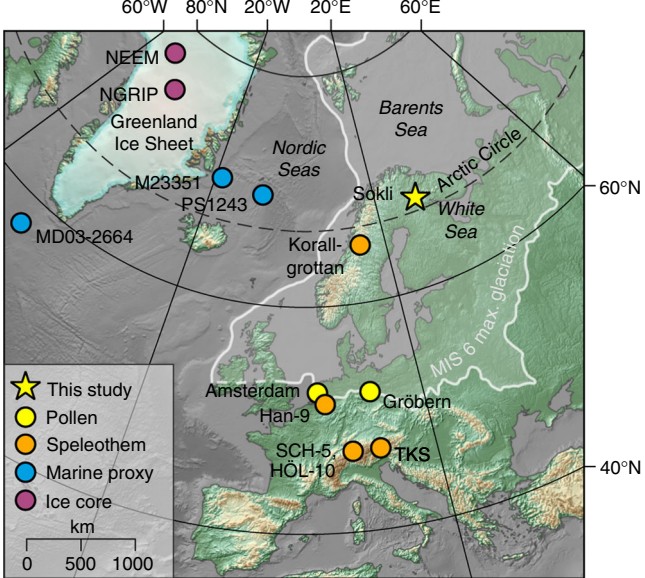

**Fig. 1** Location of Sokli and key sites referenced in the study. Maximum extent of glaciation during the Saalian ice age (Marine Isotope Stage 6)[13] is shown

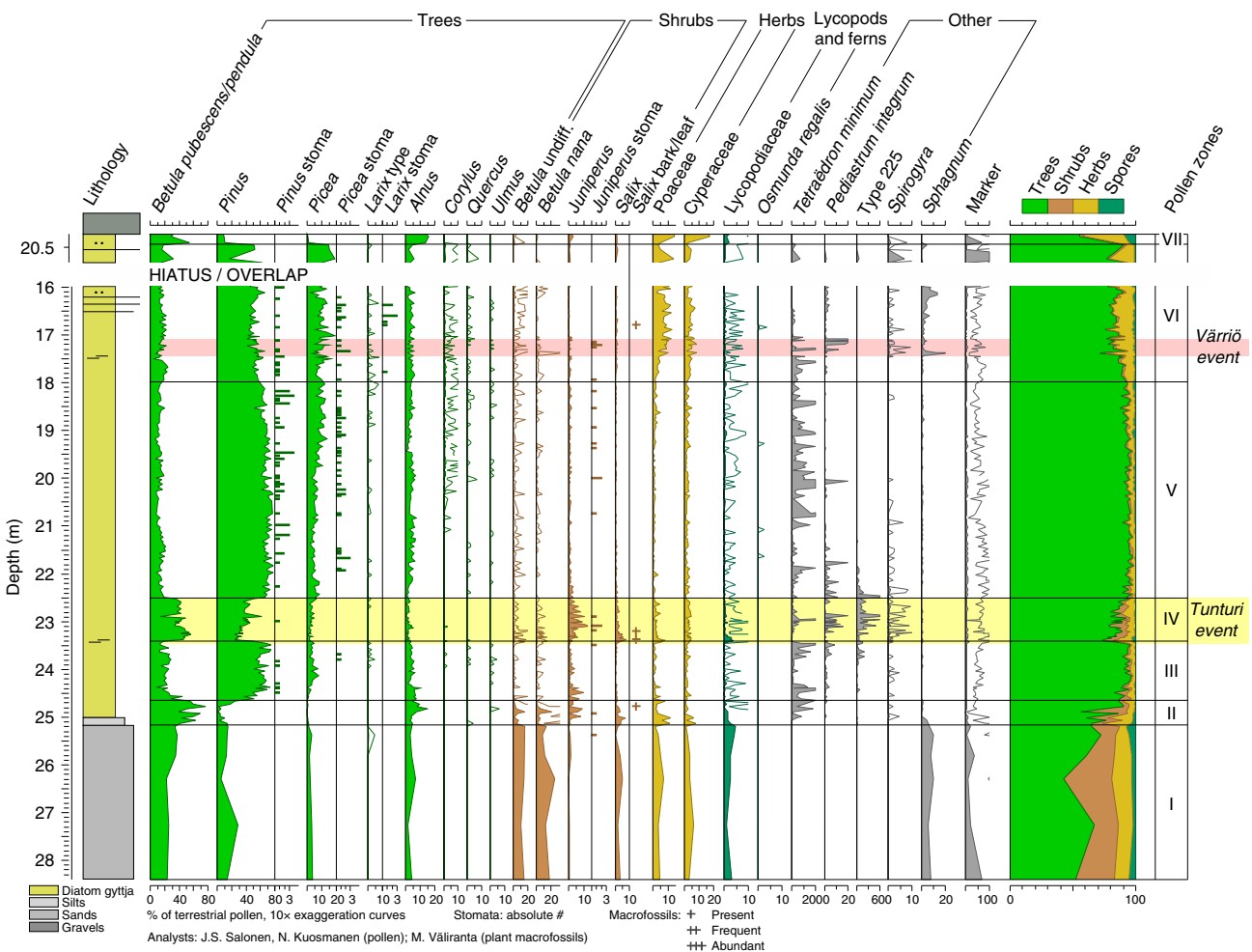

**Fig. 2** Fossil data (pollen, spores, stomata, non-pollen palynomorphs, and plant macrofossils) from the Eemian sequence at Sokli. The end of the Eemian is recorded in a parallel core, shown on top. The parallel core is shown with 3× exaggeration of the depth scale. Abrupt climate events are indicated with coloured bars (yellow = Tunturi event, pink = Värriö event). Lithology is from ref.[16]. Selected taxa are shown. The full pollen and macrofossil data and the multivariate regression tree defining the pollen zonation are shown in Supplementary Figs. 1–3

the zone, followed by a gradual decline at the expense of spruce. In Zone VI (17.98–15.98 m and 20.60–20.47 m in parallel core) the forest composition remains generally similar, however with gradual increases in Poaceae and Cyperaceae pollen, and overall increased values for *B. pubsescens/pendula*, indicating the start of the Late-Eemian climatic cooling. Noteworthy about Zone VI are the numerous finds of both pollen and stomata of *Larix* indicating its local presence. Larch is absent from modern-day Finland, with nearest natural occurrences of *L. sibirica* found at the shores of the White Sea in NW Russia[26]. In Zone VII (20.47–20.38 m in parallel core), the boreal forest is replaced by birch forest in response to continued cooling.

Between depths 17.40–17.05 m, i.e., within Zone VI, shifts are recorded in the examined proxies. This abrupt cooling and drying event is here defined the Värriö event. This event has a relatively muted signal in the terrestrial pollen record, however the initial peak (two samples at 17.37–17.39 m) is seen as a sharp minimum in total tree pollen, with the ratio of arboreal to non-arboreal pollen (mean = 2.9) below the lowest value reached during the Tunturi event (4.8) indicating the opening of tree cover, while a maximum in *Sphagnum* suggests the extension of the wetland zone. A more distinct and longer lasting signal for the Värriö event is found in the aquatic proxy record (see Discussion).

**Palaeoclimate reconstruction.** The pollen-based reconstructions for $T_{jul}$ and $T_{jan}$ are presented in Fig. 3. After a noisy interval during the pioneer vegetation of Zones I–II, the $T_{jul}$ curves (Fig. 3a) show two distinct warm stages. The individual reconstructions peak at ca. 2–3 °C above present during Zones III and V, with highest $T_{jul}$ recorded during the early part of Zone V. The Tunturi event (Zone IV) is clearly seen, with an extremely rapid onset and gradual, stepwise recovery. The peak cooling of the Tunturi event (mean $T_{jul}$ during the largest dip at 23.33–23.37 m vs. mean $T_{jul}$ during preceding warm interval (Zone III)) varies between methods at 2.3–3.5 °C, with a mean magnitude of 2.7 °C. During Zone VI a gradual decrease starts in $T_{jul}$. This temperature decline is further interrupted by the Värriö event, where the abrupt onset represents a distinct fall in the median $T_{jul}$ reconstruction during the otherwise stable warm interval following the Tunturi event. During the initial sharp cooling peak (17.37–17.39 m) the individual reconstructions show a temperature fall of 0.7–2.7 °C compared to the preceding half-metre interval, with a mean magnitude of 1.8 °C, followed by a recovery to near pre-event temperatures. Zone VII sees a return to pre-interglacial $T_{jul}$ values.

The $T_{jan}$ reconstruction (Fig. 3b) shows a muted variability for Zones I–IV but generally parallels the major features of the $T_{jul}$

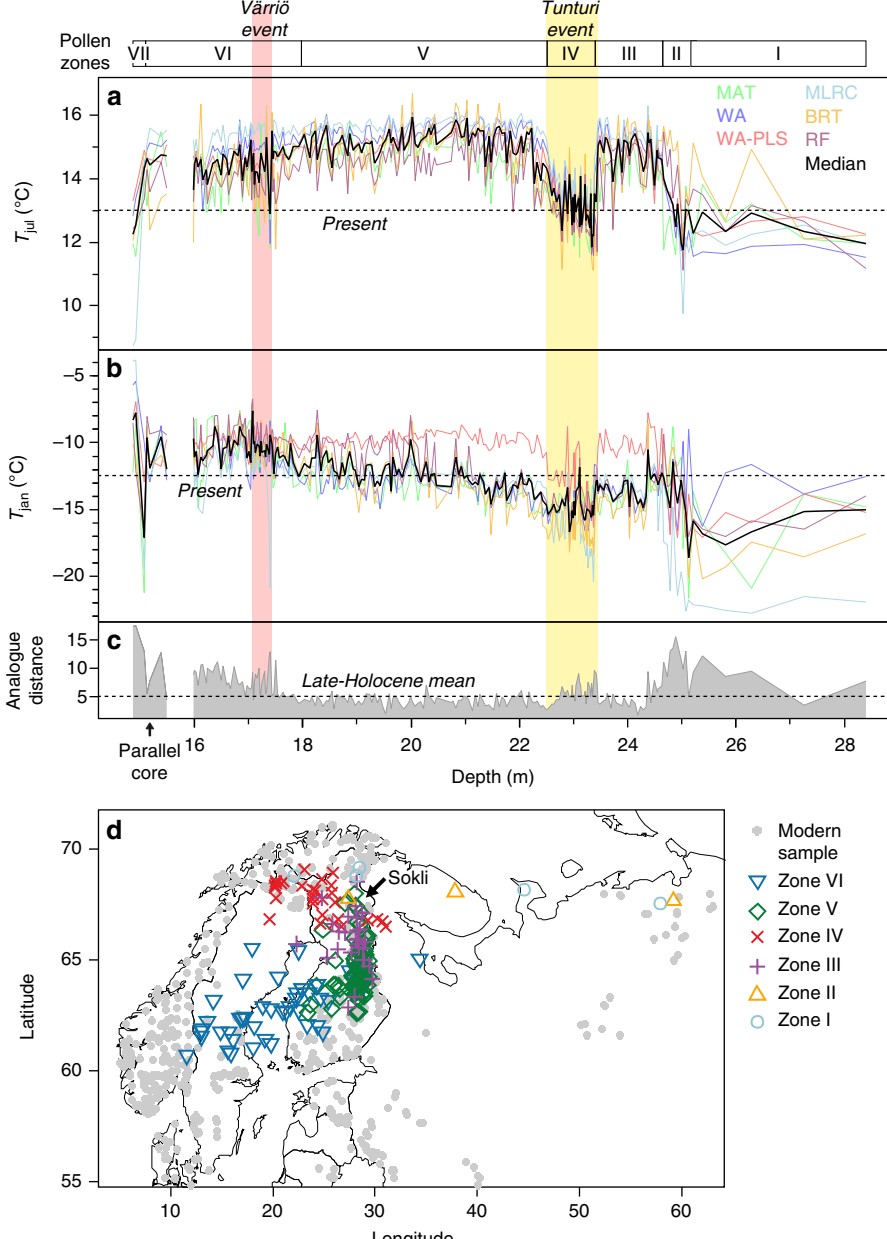

**Fig. 3** Climate reconstruction based on the Eemian pollen data from Sokli. Reconstructions are prepared for (**a**) $T_{jul}$ and (**b**) $T_{jan}$, using six different pollen–climate calibration models (see Methods for details), and summarised as a median of all six reconstructions (black curve). Abrupt climate events are indicated with coloured bars. The parallel core covering the end of the Eemian is shown with 3× exaggeration of the depth scale. **c** Analogue quality (squared chord distance[68]) between each fossil pollen sample and the best matching modern pollen calibration sample. The dashed line indicates the mean analogue quality found for Late-Holocene samples from a nearby lake core[69]. **d** Geographic locations of best modern analogues found for each fossil pollen sample. Fossil samples use symbols based on their pollen zone, and are positioned at the mean latitude and longitude (weighted by the analogue quality) of the five most compositionally similar modern pollen samples. For clarity, only fossil samples with relatively good modern analogues are shown (compositional distance to best modern analogue less than 2× the mean analogue distance found for Late-Holocene samples; see panel **c**). Locations of the available modern samples and the Sokli fossil site are also indicated

reconstruction. However in Zones V–VI, $T_{jan}$ takes a distinct, opposite trend compared to $T_{jul}$, with all but one reconstruction showing a gradual rise in $T_{jan}$ by ca. 5 °C and ending at 2–3 °C above present.

The modern pollen analogue distances for fossil pollen samples are short for the warm stages of the Eemian, being similar or lower compared to the mean distance for Late Holocene samples (Fig. 3c). This suggests a robust reconstruction with small errors due to non-analogue palaeoclimate or climate–vegetation

disequilibrium. The lack of wetland elements in the microfossil record, and the general scarcity of macrofossils, suggest that the coring site was situated in the central part of a lake. This is likely to help the pollen-based reconstruction as there is little non-climatic noise caused by local azonal vegetation, and the sequence rather samples the atmospherically mixed pollen of regional vegetation. Thus strong pollen analogues are found in the lake-based modern calibration data. However, the pollen analogue distances increase during the Early-Eemian pioneer vegetation

stage and during the abrupt events characterised by rapid species turnover. This is reflected as noise and poor between method robustness in reconstructed temperature within these depth intervals.

The geographic locations of the modern pollen analogues are plotted in Fig. 3d. For Zones I–II, characterised by semi-open pioneer *Betula* vegetation (Fig. 2) and generally large analogue distances (Fig. 3c), the best modern analogues are found from various sectors of the modern forest–tundra ecotone of NE Europe. During the first warm stage (Zone III), best analogues are located in north-central Finland, south of Sokli, while a distinct shift towards northwest occurs during the Tunturi event (zone IV). Zone V reveals a shift in the analogue locations from south-central Finland to the southwest, which continues during zone VI when the best analogues end up in central Sweden. This parallels the trend towards warmer winters reconstructed with most methods (Fig. 3b).

In significance testing using a redundancy analysis permutation test[27] (see Methods for details), the $T_{jul}$ reconstruction is significant ($p = 0.001$), while for the $T_{jan}$ reconstruction a significant result ($p = 0.011$) is yielded for pollen Zones V and VI. This covers the long and stable $T_{jan}$ increase across the upper ~60% of the sequence, but leaves out the relatively noisy $T_{jan}$ reconstruction from Zones I–IV (Fig. 3b), possibly biased by the poor modern pollen analogues (Fig. 3c).

**Age model.** To compare our results with regional palaeoclimate datasets and climate forcing time series (Fig. 4), we assign an absolute time scale to the Sokli sequence. The age model is constrained by an event-stratigraphic correlation to U/Th-dated speleothem records from Sweden, Belgium and the Northern Alps[28–30], and further supported by Greenland ice cores[31,32] and Norwegian Sea marine data[33] (see Methods for further details). Our age model suggests start of interglacial warm conditions at ca. 130 ka and end of the interglacial at ca. 117.5 ka.

## Discussion

The outstanding features of our climate reconstructions are the distinct decoupled trends in summer (falling) and winter (rising) temperature across the Eemian, and the two abrupt events in the summer temperature reconstruction. Major features of these pollen-based reconstructions are supported by the aquatic proxy record from this sequence, including green microalgae (this study) and diatoms (ref.[24]).

The diatom data corroborate the trend of increasing $T_{jan}$ reconstructed from pollen. Changes in the diatom composition reflect a prolongation of the open-water period in early Zone V, mainly as an effect of milder winters and an earlier ice-out. Following this, decreasing continentality is further indicated by decreasing stability of summer stratification; this interpretation is however complicated by decreasing water depths (due to lake infilling) which has a similar effect on the stability of stratification[24]. The Late-Eemian rise in reconstructed $T_{jan}$ also coincides with occurrences of spores of *Osmunda regalis* (Fig. 2). In modern northern Europe, *O. regalis* has a distinctly oceanic distribution, with the most continental occurrences found in southern Sweden. Thus the Late-Eemian fossil occurrences at Sokli support the reconstructed, significantly warmer winters during this time. The heavy spores of *O. regalis* are less likely to be transported by wind compared to pollen, however long-distance transport cannot be ruled out[34]. Although the $T_{jan}$ reconstruction should be interpreted with caution, due to the comparatively weak signal of winter temperature in northern European pollen data (see Methods), here we regard the first-order trend towards warmer winters, validated by independent data, as the salient feature.

The two cooling events in the pollen-based reconstructions are also seen in the aquatic record, which also provides indications of dry climate during these intervals. The occurrences of the green algae *Pediastrum integrum* and *Tetraëdron minimum*, and algal-type T.225[35], at the end of Zone III (Fig. 2) suggest that cooling of

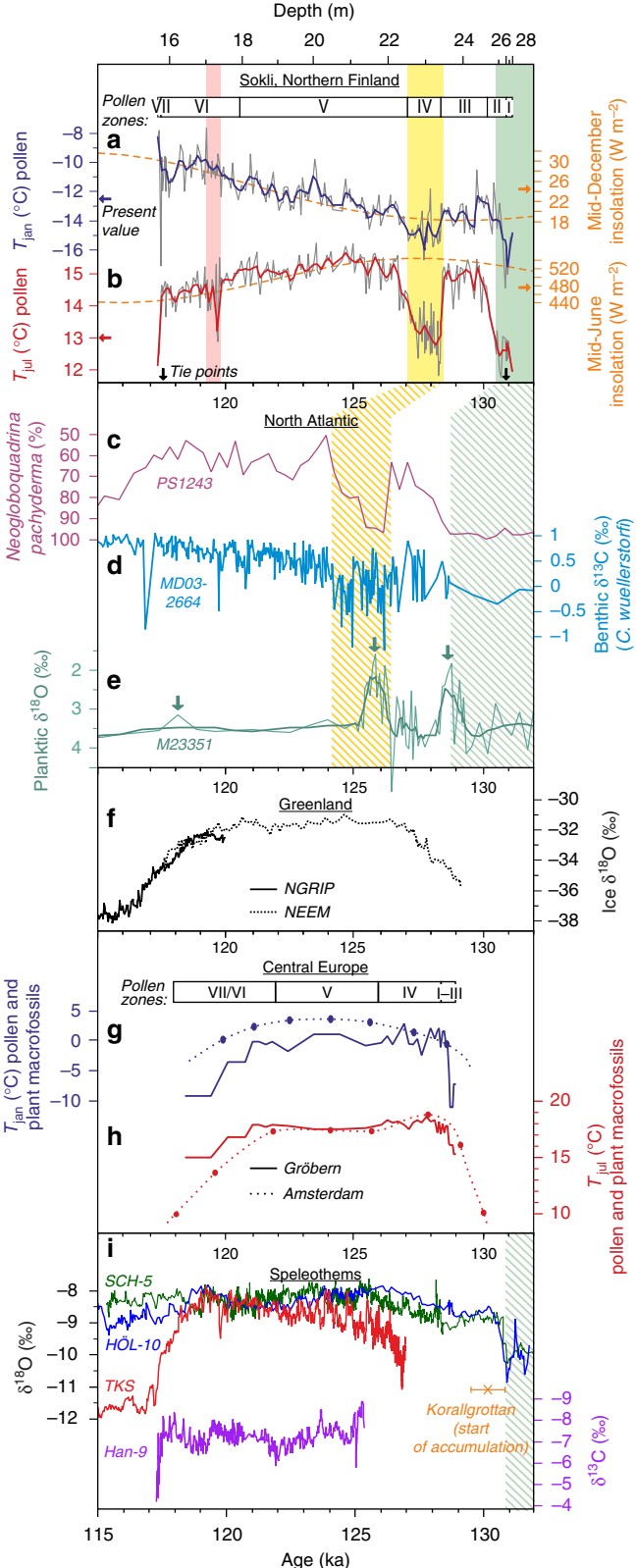

the Tunturi event may have started before the strong response in the terrestrial record (from ca. 23.8 m)[24]. Based on the combined multiproxy data, the Tunturi event appears to have two separate cold peaks, with sharp $T_{jul}$ minima at the onset and about two thirds into the event (Fig. 3a). The terrestrial fossil sequence (Fig. 2) shows a sharp recovery phase halfway through the event, with the pollen of *Pinus* increasing at the expense of *B. pubsescens/pendula*, and with a *Pinus* stoma indicating local presence. The diatom data also suggest a mid-event recovery including warming and a rise in lake level[24]. Juniper is known to be frost-susceptible and prefers a winter snow-cover[36], and its decreased values during the cooling peaks thus suggest relatively dry climate. This is further suggested by reduced values of *Alnus* pollen, expansion of the wetland zone (peak in *Salix*) and appearances of the shallow freshwater algae *Spirogyra* (Fig. 2) and of aerophilic and halophilic diatoms[24]. While the Värriö event is seen as a sharp initial peak in the pollen-based reconstructions (Fig. 3a), in the aquatic proxies the Värriö event has a longer signature, and is even more distinct than the Tunturi event. Most importantly, peak values of *Spirogyra* green algae and littoral diatom taxa indicate sudden drops in lake level during peak cooling[24], while the cold-oligotrophic green algae *P. integrum* has a strong maximum (Fig. 2), paralleling the response during the Tunturi event.

An increasing body of data from the North Atlantic and the Nordic Seas shows prominent and widespread cold and fresh-water events intersecting interglacial conditions during the Eemian[7,15,17–20]. Sea-surface temperature (SST) proxies from the Nordic Seas show a prominent cooling at ca. 125–126 ka (Fig. 4c). Records from the Eirik Drift, south of Greenland, show a con-current cooling[17], together with reductions in North Atlantic Deep Water formation, inferred from low benthic $\delta^{13}$C excursions, suggesting a weakening of the Atlantic Meridional

Overturning Circulation (AMOC; Fig. 4d)[15]. When comparing against the SST (Fig. 4c) and AMOC (Fig. 4d) proxies, the Tunturi event at Sokli (Fig. 4b) shows a similar duration and positioning between two warm periods with interglacial conditions, as well as a possible two-pronged structure of the cooling phase. At the East Greenland margin, two major glacial meltwater events have been identified at 128.5 and 126.5 ka, based on negative anomalies in planktic foraminifera $\delta^{18}$O (Fig. 4e), with Greenland Ice Sheet (GIS) melting implicated as the cause[19]. The early warm period seen at Sokli (pollen zone III) would thus coincide with the period of high SST's (Fig. 4c) and relatively strong AMOC (Fig. 4d) between the successive meltwater out-bursts. Based on alignment of the East Greenland margin and Eirik Drift records, it has been speculated[19] that a further melt-water outburst from the Laurentide Ice Sheet (LIS)[37] may have prolonged the event initiated by the 126.5 ka meltwater input from the GIS. At the East Greenland Margin, a further meltwater spike has been documented in the Late Eemian (ca. 118 ka; Fig. 4e)[19], while events of increasingly strong southward incursion of Arctic waters are seen in the subpolar North Atlantic over the Mid-to-Late Eemian[38]. Continued Late-Eemian events are also seen in the Eirik Drift proxy data (e.g., 117 and 120 ka; Fig. 4d)[15,18], possibly due to the effects of warm temperatures and the hydrological cycle on surface ocean buoyancy[15]. However due to lack of absolute dating and the general ambiguity of chronological tie points within the Eemian[10], we cannot confidently correlate the Late-Eemian Värriö event with a specific marine event.

The numerous continental Eemian sequences from the mid-latitudes of Europe do not reveal a comparable sequence of abrupt climate events, although increasing instabilities are seen in some records towards the Late Eemian[39,40], but with no distinct, consistently recorded events[39,41–43]. Isolated pollen and speleothem sequences from mid-latitude Europe[30,44–46] have however suggested abrupt events within the Eemian. Transient ocean–atmosphere models of the Eemian do not reproduce comparable short-term climatic shifts, however these modelling efforts have focused on long-term trends and have not included possible forcings of abrupt climate events seen in the proxy data, such as meltwater pulses occurring after the major Early-Eemian deglaciation[4,47].

In contrast with the mid-latitude data, the Eemian abrupt events have a strong and repeated expression in the emerging data from the high latitudes, including Sokli and the North Atlantic marine data, suggesting a northern distribution of the events. However, it is uncertain to what extent this distribution is affected by, e.g., resolution of datasets or differences in the seasonality of the events and the seasonal sensitivities of proxies in individual regions. We note that while the Eemian cooling events have proxy expressions lasting up to 2 ka (Fig. 4), they are both in our data and in the marine data[7,15] characterised by sharp and strong maxima against a longer event complex. Given the variable resilience and possible threshold responses in individual climate proxies, it might thus require a high-resolution multi-proxy dataset to resolve the events. An illustration of these challenges is provided by the Värriö event in the Sokli data. Here, only the initial sharp maximum is resolved in the pollen record, possibly because the site was located further from the forest–tundra eco-tone than during the well-resolved Tunturi event. Meanwhile the Värriö event has a longer and stronger expression in the aquatic proxies[24], likely due to already shallow conditions making the proxies sensitive to further shallowing. While the geographic and seasonal expression and the involved forcings and feedbacks remain to be fully resolved, in the context where the future evolution of the AMOC is poorly constrained (~0 to ~50% reduction by 2100 CE[48]), but with indications of a recent

---

**Fig. 4** Sokli data compared against other proxy and forcing time series. The coloured bars indicate intervals of cool climate at Sokli (solid bars) and their suggested counterparts in other datasets (hatched bars). **a**, **b** Reconstructions for $T_{jan}$ and $T_{jul}$ based on the Sokli pollen sequence. The reconstructions are expressed as the multi-method median curves (grey curve) with local regression (LOESS; span 0.03, one robustifying iteration) smoothers (red and blue curves) fitted. Black arrows indicate the tie points of the absolute chronology used for Sokli. December and June insolation at 60°N[5] are also shown (orange curves). **c–e** Marine proxy data from the Nordic and Labrador Seas, including sea-surface temperature proxy data (% *Neogloboquadrina pachyderma*) from the central Nordic Seas[19,70] (**c**), $\delta^{13}$C proxy data for North Atlantic Deep Water formation[15] (**d**), and planktic $\delta^{18}$O data from East Greenland Margin[19] (**e**). In **e**, green arrows indicate interpreted locations of glacial meltwater pulses. All marine data are shown aligned[19] with the AICC2012 ice-core chronology[71,72]. **f** Ice $\delta^{18}$O records from NEEM[31] and NGRIP[32] ice cores from Greenland, displayed on the AICC2012 chronology[10,71,72]. **g**, **h** Pollen and macrofossil based reconstructions for $T_{jan}$ and $T_{jul}$ from the European mid-latitudes. Reconstructions are shown based on the Gröbern (Germany) fossil sequence[41] and for Amsterdam (Netherlands) interpolated from Eemian palaeo-isotherms reconstructed based on fossil data from multiple sites (adapted from ref.[50]). The time scales used are floating chronologies based on estimated pollen zone durations[41], shown here anchored at 129 ka. Pollen zones are shown for Central Germany[41]. **i** Speleothem stable isotope data underlying the chronology for Sokli, including $\delta^{18}$O data for SCH-5 and HÖL-10 speleothems from the Northern Alps[28] and the TKS series from the Entrische Kirche cave in the Eastern Alps[46], as well as $\delta^{13}$C data for the Han-9 speleothem from Belgium[29]. The cross and an error bar indicate the age and associated 2σ uncertainty for the onset of speleothem growth at Korallgrottan, northern Sweden

slowdown[49], the Eemian abrupt events provide compelling evidence of repeated, realised instabilities against warmer than preindustrial background conditions[15,19]. Importantly, while the Early-Eemian freshwater and cold events (up to ca. 125 ka) may be tied to the main Saalian deglaciation and thus may not represent purely interglacial boundary conditions[19], the Värriö event and the Mid/Late-Eemian abrupt events identified in the Labrador[15,17] and Nordic Seas[19] show that the instabilities persisted under warm interglacial conditions.

In long-term Eemian climate trends, summer temperature reconstructions from central and western Europe consistently show an Early-Eemian maximum and a following cooling[40,42,43,50] (Fig. 4h). With the exception of the abrupt events, the first-order trend in our $T_{jul}$ reconstruction (Fig. 4b) agrees with these findings. For winter temperature (Fig. 4g), the mid-latitude European reconstructions also generally show a falling trend[39–43]. Some disagreement concerns whether the distinct shift towards colder winters happens already at the start of the Mid Eemian[39,40,42] or later, over the Late Eemian[41,43,50]. These discrepancies have been attributed to the abundant *Carpinus* pollen in Mid-Eemian deposits (pollen Zone V in Fig. 4g, h) and human impact or competition affecting the modern distribution of this taxon, possibly biasing the pollen–climate modelling underpinning palaeo-reconstructions[39,41]. In contrast with these data from the mid-latitudes, our data show a long rise of $T_{jan}$ persisting through the Mid and Late Eemian, and generally running opposite to the falling $T_{jul}$ trend. This follows the secular insolation forcing in the northern high latitudes (Fig. 4a, b).

In an ensemble of transient climate models of the Eemian[4], winter temperatures at 60–90 °N tend to fall in sync with summer insolation, due to the effect of summer temperature on sea ice formation during the following winter, thus locking winter temperature evolution with the summer forcing. By contrast, in northern mid-latitudes (30–60 °N) the modelled $T_{jan}$ robustly follows the rising trend in winter insolation. Our study area is located roughly where the transition in the winter trend occurs[51], suggesting that cryosphere feedbacks should be pivotal in shaping winter climate in the region. However, current climate models have a generally poor robustness in regions affected by Arctic cryosphere feedbacks[4,8,51], underlining the need for high-latitude proxy data with distinct winter climate signals to constrain the modelling.

Our data, showing a persistent $T_{jan}$ rise, suggest a limited effect of summer insolation and sea-ice feedbacks in driving Eemian winter temperature evolution in the European Arctic. While the $T_{jan}$ rise parallels the trend in winter insolation forcing, the magnitude of winter warming is surprising considering the small absolute variation of winter insolation at high latitudes[4] (Fig. 4a). This discrepancy may be explained by the suggested increasingly vigorous AMOC[15,52] (Fig. 4d), and proximally by the warm Nordic Seas[7,19,20,52] in the Late Eemian. Late-Eemian maxima in AMOC strength[4,47] and Nordic Seas surface temperature[52,53] are also predicted by transient modelling when including prescribed Northern Hemisphere ice-sheet dynamics, due to an uptake of freshwater by the expanding ice sheets, with the first-order changes in modelled AMOC matching the proxy data[4,15]. A strong AMOC and oceanic heat transport would also help explain the apparent contradiction of warmer-than-present summers persisting into the Late Eemian, despite the low insolation (Fig. 4b). The winter warmth could be further induced by the strong westerlies suggested to have prevailed in the Late-Eemian northern Europe[52]. In climate modelling, strong westerlies and localised sea-ice feedbacks in the Barents Sea have been found to produce a strong $T_{jan}$ anomaly (+4–6 °C) centred over northeast Europe while only a minor anomaly (+0–2 °C) reaches the central and western European Eemian data sites[12], a pattern

consistent with the deviating Late-Eemian winter climate evolution seen at Sokli. The Late-Eemian climate at Sokli lends support to the hypothesis[38,52] of last glacial inception and nucleation of the Scandinavian ice sheet starting due to enhanced AMOC combined with low summer insolation and related feedbacks, resulting in increased winter precipitation and reduced summer snow melt. The cool summers but warm winters reconstructed at Sokli, suggesting a strong oceanic influence and increased precipitation, show climate conditions consistent with this hypothesis on the northern Fennoscandian continent, proximal to the site of glacial inception.

The Eemian climate evolution in northernmost continental Europe reveals a distinct mixed influence of oceanic and insolation forcings, producing a decoupled evolution of winter and summer climate which in Europe appears to be unique to the high latitudes. These long-term climate trends are further modified by repeated abrupt cooling events persisting through the Eemian, linked to disturbances in the North Atlantic circulation regime.

## Methods

**Site.** The Sokli basin in north-east Finland (67°48′ N 29°18′ E, 220 m a.s.l.) is one of the few terrestrial sites in northern Europe where Eemian sediments have been found preserved in a stratigraphic sequence, with overlying glacial till beds and non-glacial sediments of Weichselian age. The Sokli sequence has escaped major glacial erosion due to non-typical bedrock conditions. The Eemian diatom gyttja deposit at Sokli stretches as a marker horizon near the base of the unconsolidated sediment infill. Its interglacial pollen content was first noted in the 1970s and was correlated with the Eemian. Detailed stratigraphic studies on the overlying Weichselian sediment sequence, combined with absolute dating control, have supported the Eemian age assignment[11,21,22]. The diatom gyttja bed is bracketed by thermoluminescence and infrared stimulated luminescence dating to >ca. 110 and <ca. 150–180 kyr[21], and by optically stimulated luminescence dating on quartz (using the single-aliquot regeneration dose protocol) to >ca. 95 kyr[22,54].

**Coring and sampling.** The present study was conducted on a new borehole Sokli 2010/4 that was cored at a site located between boreholes Sokli A/B-series and 900/901 from the central part of the Sokli basin and 902/905 from the basin margin[21,22]. Additionally, several samples from core 901 were re-analysed for pollen[21]. The latter core records the final birch phase that is missing (truncated) at the site of borehole Sokli 2010/4.

**Proxy analyses.** Pollen samples were prepared from 1 cm³ subsamples, using HCl, KOH, sieving (212 µm mesh), Na-pyrophosphate, acetolysis, and bromoform heavy-liquid treatments, and mounted in glycerol. *Eucalyptus* markers were added to estimate absolute pollen concentrations. A mean of 401 (min = 220.5, max = 526) terrestrial pollen and spore grains were counted from each sample. Pollen percentages were calculated from the sum of all terrestrial pollen and spores. Conifer stomata, charcoal, marker grains, and non-pollen palynomorphs were also counted from the pollen slides. A total of 217 pollen samples were counted, mostly at 4 or 6 cm intervals but going down to 2 cm in sections with rapid species turnover, providing a generally sub-centennial resolution across the Eemian.

A pollen zonation was calculated with a multivariate regression tree[55], implemented using the MVPART library[56] (version 1.6-1) for R. Fossil assemblages for terrestrial pollen and spores were used as multivariate response and core depth as the sole predictor. The tree size was determined by cross-validation, using the smallest tree within one standard error of the best tree.

Plant macrofossil samples were prepared at 4–20 cm intervals from subsamples of mainly ca. 5 cm³. The sediment was sieved using a 100 µm mesh under running water and the residue examined using stereo and high-magnification light microscopes.

**Climate reconstruction.** Climate reconstructions for $T_{jul}$ and $T_{jan}$ were prepared for the fossil pollen samples using all terrestrial pollen and spore taxa. An ensemble of pollen–climate calibration models (Supplementary Table 1) were used, including regression tree-based machine-learning approaches (boosted regression tree (BRT), random forest (RF)), unimodal multivariate transfer functions (weighted averaging (WA), weighted averaging-partial least squares (WA-PLS), maximum likelihood response surfaces (MLRC)), and the matching of fossil samples with most similar modern samples (modern-analogue technique (MAT)). The calibration models were built on a set of 807 surface pollen samples from Europe, with modern climate data extracted for each sample. The climate reconstructions were extensively validated, including calibration model cross-validations (Supplementary Table 1), analysis of the quality of modern analogues found for the fossil pollen samples (Fig. 3c), analysis of model structure and comparison against ecological knowledge

(Supplementary Tables 2 and 3), testing the statistical significance of the palaeo-reconstructions, and validation against palaeoclimate inferences from independent aquatic proxy data. For a full description of the reconstruction method, see Supplementary Methods.

**Chronology**. We use an age of 130.9 ± 1 ka (2σ) for the onset of the Eemian based on correlation to the base of an abrupt warming event recorded in speleothem δ[18]O records from the Northern Alps (SCH-5 and HÖL-10; Fig. 4i)[28]. This event is inferred to represent the initiation of the interglacial in the Alps, where interglacial conditions were established at latest by ~130 ka[28]. This age is earlier than the start of the last interglacial when inferred from an abrupt intensification of the Asian Monsoon recorded by speleothem δ[18]O records from Sanbao Cave in China[57] and linked to the North Atlantic warming following the Heinrich Event 11, but coincides, within the chronological uncertainties, with a smaller monsoon intensification event[28]. Although there will always remain ambiguities in such event-stratigraphic correlations, we have chosen to use the event in the Alpine speleothem records as the starting point for the Eemian in Sokli. This correlation is supported by recent U/Th dating of a stalagmite from the Korallgrottan (Coral Cave) in northern Sweden, indicating deglaciation of central parts of the northern Fennoscandian Ice Sheet and onset of stalagmite growth by 130.17 ± 0.66 ka (Fig. 4i) (Personal communication: Frank, N. Institute of Environmental Physics, Heidelberg, Germany). An early deglaciation of the Fennoscandian Ice Sheet is also inferred from a speleothem record from the Okshola cave in Northern Norway[58]. The latter record is, however, dated by alpha-spectrometry having a comparatively low chronological precision. We argue that the Sokli site became deglaciated earlier than the Korallgrottan site during the penultimate deglaciation, by analogy to the last deglaciation when the Sokli site is estimated to have become ice free about 1 ka before the Korallgrottan[59,60]. Based on this phasing and the dating from Korallgrottan, we analogously infer that Sokli had been deglaciated at the time of the Alpine warming starting at 130.9 ka. We therefore use this date for the transition into the Eemian at Sokli.

For the end of the Eemian we use an age of 117.5 ± 0.5 ka (2σ), based on U/Th dating of several speleothem records from Belgium indicating establishment of cold and dry conditions, suggesting end of interglacial conditions in Northern Europe (Han-9; Fig. 4i)[29,30]. Similarly, a major 3% drop in calcite δ[18]O recorded in a speleothem record from Entrische Kirche cave in Austria[46] indicates a decline in temperatures along with other climatic changes at ca. 118 ka (TKS; Fig. 4i). This abrupt cooling is, however, not as prominent in other Alpine speleothem δ[18]O records[28]. A concurrent end of interglacial temperatures in the high latitude North Atlantic region is further suggested by δ[18]O data from the NGRIP and NEEM ice cores[31,32] (Fig. 4f) that has been linked to a shift in the ocean circulation interpreted from proxy records in a core (MD99-2289) from the eastern Norwegian Sea[33]. Here, the arrival of ice rafted tephra from Jan Mayen and an abrupt drop in the sediment calcium content suggest that the strong Eemian inflow of Atlantic Water via the Norwegian Atlantic Current came to an end, and that the eastern Norwegian Sea became dominated by Arctic Water. This event may also correlate with a first ice-rafted detritus pulse recorded at the Eirik Drift south of Greenland[18]. With glacial-like climatic conditions thus established in the eastern Norwegian Sea[33], we find it very unlikely that warmer than present summer temperatures would have prevailed in Northern Fennoscandia. Earlier work on Sokli[16] used a tentative Eemian chronology based on alignment of the pollen-based $T_{jul}$ curve with a SST record from the Norwegian Sea[7]. However with the availability of local absolute dating for Northern Fennoscandian deglaciation, this chronology is revised here.

In the Sokli sequence, the beginning of forest development at the base of pollen Zone II marks the onset of interglacial conditions, while the decline is placed where the boreal forest is replaced by birch forest at the base of pollen Zone VII (Fig. 2). We thus assign the ages of 130.9 ± 1 ka (2σ) and 117.5 ± 0.5 ka (2σ) to the bases of Zones II and VII, respectively.

To estimate the age uncertainties within the Eemian, we model the sediment deposition as a Poisson process (the P_Sequence option) using Oxcal 4.3[61–63]. In the P_Sequence option, the parameter, $k$, needs to be specified as it defines the size of each depositional event and therefore how variable and uncertain the sedimentation is. It is possible for Oxcal to estimate the $k$ by using the variable $k$ option[63], however having only two age control points, this is impossible for the Eemian sequence. We therefore apply a $k$ estimated by the variable $k$ function from a [14]C-dated lake sequence deposited in the Sokli basin during the Holocene, showing a moderate variation in deposition rate[64]. By contrast, deposition during the Eemian appears generally stable, based on the homogenous lithology (diatom gyttja) and the stable sediment pollen concentration (represented by the marker grain fraction; Fig. 2) which during a period of consistent vegetation structure (boreal forest) suggests a stable sedimentation rate. We therefore regard the age uncertainty estimates of the Eemian Sokli record as conservative. The chronological errors suggested by the age–depth model (Supplementary Fig. 4) are largest in the Mid Eemian, increasing with distance from the tie points, reaching 2σ values of ~1500 a.

The silty ice-lake beds below tie point 1 likely represent a shorter time interval than suggested by a linear extrapolation. Following the later MIS 2 and MIS 4 glaciations at Sokli, the durations of glacio-lacustrine sedimentation are estimated at less than 100 years[65,66] and up to ca. 400 years[64,67]. Based on these analogues, we

show the glacio-lacustrine section (Zone I) as representing 250 years in Fig. 4. In the age–depth modelling the parallel core covering the end of the Eemian is assumed a continuous extension of the main sequence. Some hiatus or overlap is likely, however, there is no clear indication of either in the microfossil content. The assumed continuity is supported by the pollen zonation, which finds no zone boundary at the base of the parallel core, but Zone VI runs from the main sequence into the parallel core.

**Code availability**. R code related to this paper are available on figshare (https://doi.org/10.6084/m9.figshare.6490424).

**Data availability**. The fossil data and the climate reconstructions are available on figshare (https://doi.org/10.6084/m9.figshare.6490442).

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

## Acknowledgements

J.S.S. acknowledges funding from Academy of Finland (projects 1278692 and 1310649) and the Finnish Cultural Foundation, K.H. from the Swedish Nuclear Fuel and Waste Management Company (SKB) and the Bolin Centre for Climate Research at Stockholm University, Sweden, and J.B. from the Research Council of Norway through grant no. 221999 and from the Bergen Research Foundation through the project "Earth System Interactions and Information Transfer". We thank Emilie Capron, Aline Govin, Michael Meyer, and Anastasia Zhuravleva for their help in acquiring data for this study, and N. Frank (Institute of Environmental Physics, Heidelberg, Germany) for the Korallgrottan stalagmite dating results.

## Author contributions

J.S.S. and K.F.H. had the main responsibility in designing the study and writing the manuscript. J.S.S., M.V., N.K., M.Ky. and A.Ph. prepared the proxy data analyses. J.S.S., S.G., M.Ko. and M.L. prepared the palaeoclimate reconstructions. J.B., J.S.S., A.Pl. and K.H. prepared the age model. J.B., H.R. and A.Pl. provided expertise for the

data–modelling synthesis. All authors participated in analysing the results and contributed comments.

## Additional information

**Competing interests:** The authors declare no competing interests.

