## [Peer Review File · Nature Communications]

Reviewers' comments:

Reviewer #1 (Remarks to the Author):

"Abrupt high-latitude climate events and decoupled seasonal trends during the Eemian Interglacial" Salonen et al.

Salonen et al reconstruct Eemian climate from the important Sokli location in Northern Finland. The manuscript is well written.

The summer temperature reconstruction seems reasonable, and events in the reconstruction can be related to climate events in from other proxy records. Although the authors have done a good job to validate their ability to reconstruct winter temperature, certainly one of the best I have seen, I remain somewhat sceptical about the winter temperature reconstruction. How much, for example, could changes in precipitation drive the reconstruction?

Should (by analogy with the Holocene) the reconstructions be corrected for land uplift? Is the Eemian climate in northern Finland expected to have good climatic analogues in the modern climate? How might this affect the reconstructions, particularly of winter temperature?

The manuscript should state where the data and code needed to reproduce the results are to be archived.

Perhaps this is a bizarre journal policy, but separating the figures from their caption and having both remote from the text that refers to them is a certain way to frustrate and annoy the reviewer. Please do not do this.

182 The modern pollen analogue distances (Fig. 3C) for fossil pollen samples are small -> short

224-226 This sentence is contradictory.

297 global climate heading towards Eemian Interglacial levels - Climate does not have levels, temperature instead?

324 This would appear to contradict the rest of the paragraph. At 67N, one might expect that Arctic Sea ice would be important, and thus the indirect influence of summer insolation affect winter temperatures. But the reconstruction shows rising winter temperatures following winter insolation. Maybe "are pivotal" should read "should be pivotal"

412 testing the correlation and independent effect of the reconstructed variables in the calibration data - unclear. Is this with an ordination?

709 Neogloboquadrina pachyderma -> is this sinistral or dextral (or both)

Supplementary material

The significance of the partial CCA is of little importance, especially as autocorrelation in the calibration set has not been accounted for. The proportion of variance explained by each variable and their covariance would be more useful guide to the importance of each seasons temperature.

What is the ratio of the first to second eigenvalue for ordinations constrained by just summer or winter temperature (see Juggins 2013)

Reviewer #2 (Remarks to the Author):

The Last interglacial (LIG, ~129-116 ka) offers an opportunity to study the response of the Greenland and Antarctic ice sheets to a warmer-than-preindustrial high-latitude climate. While climate changes above the polar ice sheets and at the surface of the oceans across the LIG have been increasingly documented over the past years, there is still a lack of high-resolution paleodata informing on climate changes in land-areas, especially in the high latitude regions. This hampers our ability to fully understand the climate feedbacks as well as the processes associated to land-sea contrast that occur in those regions. The high latitudes are particularly sensitive to changes in the radiative forcing and act as amplifiers of climate change. Studies presenting new insights on high-latitude terrestrial climate changes across the LIG are thus needed.

In this context, the study of J. Sakari Salonen and co-authors is very relevant and will be of great interest for the paleoclimate community since it presents seasonal temperature reconstructions from the Sokli lacustrine sediment record from Northern Finland using pollen-climate calibration models. They emphasize that their climate reconstructions (January and July temperatures) suggests 1) the existence of two abrupt cooling events in this region and 2) decoupled trends between the summer temperature evolution and the winter temperature evolution across the LIG.

Overall the authors have developed an interesting study in a well-written paper accompanied with sound explanations of the methodology and appropriate figures. However I have a doubt that enough evidences are provided for its conclusions and that the claims are novel enough to guaranty its publications in Nature Communications.

Indeed, a large part of their manuscript is dedicated to the description and discussion of the two abrupt cooling events referred to as the Värriö and Tunturi events. As interesting as the identification of these events is, they have already been described in the Sokli record (based on other proxies) in previous publications (e.g. Helmens et al. 2015, Pliik et al. 2016). Actually as far as I understand the authors refer a couple of time to the Tunturi event identified previously in the introduction in several places without naming it (line 84 and line 97).

Also, in several places in the manuscript (further described below), the proposed statements are "overselling" what can actually be seen from their reconstructions. For instance it is stated in the the abstract that "the peak interglacial conditions are intersected by two strong cooling and drying events". However (1) according to the dating of these events, we cannot rule out that those events are not happening during purely interglacial conditions e.g. The Tunturi event could be associated to the penultimate deglaciation. Also to me (2) the signature of the Värriö event in the pollen markers is neither strong nor clear and as a result neither it is in the January and July temperature reconstructions.

Thus, publishing those results in a more specialized journal might be appropriate. I detail below some important comments that the authors should consider when preparing a revised version of their manuscript.

MAJOR COMMENTS:

- Abrupt cooling events in the Sokli record
 - In this manuscript, the authors report on two cooling events that they evidence at the beginning (the Tunturi event) and at the end of the LIG (the Värriö event) in the pollen records. Both events have been previously described in studies based on other proxies (e.g. Helmens et al. 2015, Pliik et al. 2016). Based on the fossil data presented in Figure 2, I find that the imprint of the Värriö event in those dataset is not straightforward. Therefore I am uncomfortable with their statement in line 154 "...shifts are recorded in all proxies". The authors need to be more specific here about which proxies they are talking about.
 - I have a similar criticism regarding this event looking now at the seasonal temperature reconstructions. I do not think that the level of variability observed in the records allows to identify with confidence the Värriö cooling event both the median Tjan and Tjuly reconstructions. Indeed

the authors claim that the abrupt onset represent the largest Tjul fall in all reconstructions during the prolonged warm interval following the Tunturi event but how significant is this fall? On how many data point is this based?

The authors need to build a stronger a more convincing case. In particular it would be helpful to ease the reading of the results to have additional figures that display all the temperature reconstructions based on the different methods in such a way that is easy and straightforward to look at them one by one with both the raw and smoothed reconstructed temperature data.

-Regarding the Tunturi event, the authors claim also that all methods show distinct cold peaks within the Tunturi events. Again, the author need to build a stronger case if they want to be so definite that the variations that are reconstructed within the cooling event represent significant climate signals and are not simply related to the noisiness of the signal. Otherwise, the authors should toned down their statement and point out clearly the limitations of their observations.

I believe that because of the limitations of the temperature reconstructions raised above, the authors should toned down a few of their statements throughout the manuscript, for instance: Line 37: "...the peak interglacial conditions are intersected by two cooling events" (note also as discussed also in the manuscript that the early cooling event may not represent purely interglacial conditions but be tied to the preceding termination. The authors should thus not use the expression "peak interglacial").

Line 101: "Our climate reconstruction provide a robust recorded of repeated Eemian abrupt events...".

- Age model of the Sokli record

It is essential that more information is provided regarding the age model of the record. In particular I find it very problematic that the speleothem records used to define the two tie points at the onset and at the end of the LIG in the Sokli record are not shown in any of the figures. The resulting alignment of the pollen record onto the speleothem records need to be showed. The climatic assumption used to use the dates of onset and end of the LIG as seen in the Northern Alps and Belgium speleothem needs to be clearly spelt out and also discussed. In particular the authors should provide the limitations associated with their age model and also try and quantify the associated uncertainties.

- Global temperature during the LIG

Lines 32, 51, 53, 61: The authors state that the LIG was globally warmer that today. The authors need to be very careful here and rephrase their statements. I believe that they are based on the estimate of ~1.5°C higher than today provided by the Turney and Jones (2010) climate synthesis. However, this estimate is attached to very important limitations leading to a quantitative estimate of the global LIG warmth that is unreliable and possibly overestimating the global LIG warming. Indeed, this estimate is based on the assumption that the LIG peak warmth is synchronous across the world while there are multiple evidences now in paleo records that this is not correct (e.g. Govin et al. 2012, Capron et al. 2014, Hoffman et al. 2017). As a result the Turney and Jones (2010) synthesis does not represent a climate state at any specific time periods across the LIG but a virtual (and unrealistic) image on the LIG peak warmth across the globe.

The latest global SST synthesis based on coherent chronologies and allowing for asynchronous climate changes across the LIG suggest a global SST average that is indistinguishable from the 1995-2014 mean (Hoffman et al. 2017; the authors should also consider adding a reference to this paper).

However, it is safe to state (and very relevant to their study) that the high latitudes were warmer than today during the LIG as seen in polar records (e.g. Masson-Delmotte et al. 2011; Landais et al. 2016; NEEM community members 2012), as well as in the latest syntheses of marine records from the North Atlantic and Southern Ocean (Hoffman et al. 2017; Capron et al. 2014, 2017) that provide a spatio-temporal evolution of the LIG climate and are based on coherent chronologies. Thus I suggest the authors to rephrase their statements (and possibly refer to high-latitude temperature estimates instead of global temperature).

MINOR COMMENTS:

Title, lines 31 & 50: The “Eemian Interglacial period” is an awkward formulation. The authors should refer to either “the Eemian” or “the last interglacial period” (or simply “the Last Interglacial”) but not a combination of both.

Line 31: I appreciate that depending on the definition for the LIG used, the time interval can vary (as shown in Figure 1 from Govin et al. 2015). But I suggest the authors to use here the time interval “129-116 ka” given in the last IPCC report by Masson-Delmotte et al. (2013) defined based on eustatic sea level variations.

Line 54: “significantly reduced ice sheet in Greenland”; this statement needs to be hampered and the authors should cite Dutton et al. (2015) instead of Dutton and Lambeck (2012). They provide an updated assessment and summary of the LIG sea level changes and the likely respective contributions of the Greenland and Antarctic ice sheets to the sea level rise. Together with the results from the NEEM community members (2012), it seems that the contribution of Greenland to the LIG global sea level rise was more modest than previously stated older studies.

Line 61: follow my previous comment, the authors should write: “...to HIGH-LATITUDE climate warming...”

Line 74: Northern Hemisphere instead of Northern-Hemisphere.

Line 97: “multi-proxies” is very vague, the authors should be more specific here.

Line 185: “...that the coring site was situations in the central part...” this sentence needs to be revised.

Line 263: I agree that chronological uncertainties are too large to draw confident links between the Värriö event and changes in marine records at that time, but it would be worth considering to cite also to the Mokeddem et al. (2014) paper which suggests repeated cold water-mass expansions into the subpolar latitude throughout the LIG.

Line 678: Caption of Figure 2. Please be more specific about the colored bars and which events they indicated e.g. Yellow bar indicates the Tunturi Event and pink bar indicates the Värriö event.

Line 704: What are the LOESS smoothers? This needs to be clarified.

Line 713: Change sentence into “NEEM and NGRIP ice d18O records (38, 39) displayed on the the AICC2012 chronology (11, 44, 45).

Line 718: “the time scales used are floating chronologies” What do you mean by floating?

REFERENCES:

Capron et al. 2014. Temporal and spatial structure of multi-millennial temperature changes at high latitudes during the Last Interglacial, QSR, 103, 116-133.

Capron et al. 2017. Critical evaluation of climate syntheses to benchmark CMIP6/PMIP4 127 ka Last Interglacial simulations in the high-latitude regions, QSR, 168, 137-150.

Dutton et al. Science. Sea-level rise due to polar ice-sheet mass loss during past warm periods. Science, 349(6244), 2015.

Govin et al. 2015. Sequence of events from the onset to the demise of the Last Interglacial: evaluating strengths and limitations of chronologies used in climatic archives. QSR 129, 1-36.

Helmens, K.F. et al. Major cooling intersecting peak Eemian Interglacial warmth in northern Europe. Quat. Sci. Rev. 122, 293–299 (2015).

Hoffman et al. 2017. Regional and global sea-surface temperatures during the last interglaciation. Science 355, 276-279.

Landais, A., Masson-Delmotte, V., Capron, E., Langebroek, P. M., Bakker, P., Stone, E. J., Merz, N., Raible, C. C., Fischer, H., Orsi, A., Prié, F., Vinther, B., Dahl-Jensen, D., 2016. How warm was Greenland during the last interglacial period? Clim. Past, 12, 1933-1948.

<http://dx.doi.org/10.5194/cp-2016-28>.

Masson-Delmotte et al. 2011. Sensitivity of interglacial Greenland temperature and d18O: ice core data, orbital and increased CO2 climate simulations. Clim. Past 7, 1041-1059.

Masson-Delmotte et al. 2013. Climate Change 2013: the Physical Science Basis. Contribution of Working Group I to the Fifth Assessment Report of the Intergovernmental Panel on Climate Change. Cambridge University Press, Cambridge, United Kingdom and New York, NY, USA, pp. 383-464 (Chapter 5).

Mokeddem Z. et al. 2014. Oceanographic dynamics and the end of the last interglacial in the subpolar North Atlantic, PNAS, 111, 11263-11268.

NEEM community members 2012. Eemian interglacial reconstructed from a Greenland folded ice core, *Nature*, 493, 498-494.

NorthGRIP project members, 2004. High-resolution record of Northern Hemisphere climate extending into the last interglacial period. *Nature* 431, 147–151.

Pliikk, A., et al. Development of an Eemian (MIS 5e) Interglacial palaeolake at Sokli (N Finland) inferred using multiple proxies. *Palaeogeogr. Palaeoclimatol. Palaeoecol.* 463, 11–26 (2016).

Turney & Jones 2010. Does the Agulhas Current amplify global temperatures during super-interglacials? *J. Quat. Sci.* 25 (6), 839-843.

Reviewer #3 (Remarks to the Author):

Saloranta and co-authors present quantitative reconstructions of July and January mean air temperature during the Eemian interglacial, based on high-resolution pollen record from the Sokli site in northern Finland. The work is based on previously published findings (Helmens et al., 2015; Pliikk et al., 2016; Kylander et al., 2018) and the new data from the study site. The temperature reconstructions are statistically significant, derived from a number of pollen-climate calibration models and extensively validated, e.g., by testing statistical significance. The combined results presented in the manuscript (e.g., complete temperature dataset for July and January) could contribute to the understanding of interglacial climates as well as used in models. Of particular interest are the inferred decoupled trends in summer and winter temperature across the Eemian, ascribed to combined influence of insolation and oceanic forcing. Due to general paucity of available high-latitude terrestrial records and the particularly high-resolution of the Sokli sequence, the reconstructed winter temperature record is unique and could influence thinking in the field. The study also investigates the succession of climatic events on land, linking them to documented shifts in the oceanic circulation.

The paper is generally well-written, but in some aspects is too detailed. Results and interpretations appear sound, although sometimes they are mixed together. Also, the introduction should be more specific, the storyline in the discussion more focused. Some figure corrections are also essential. My suggestions to improve the manuscript are:

- 1) shorten the introduction, making it more specific. Authors should clearly state the problem they want to investigate and remove the unnecessary descriptions.
- 2) shorten the "Results". I found this chapter to be particularly dense reading, with the description of all the proxies (even those that are not used for temperature reconstructions). This part also seems to be somewhat repetitive, considering the earlier studies from the Sokli site. In the subchapter "Fossil record and vegetation changes" temperature-related proxies are mixed with sedimentological data and descriptions about wet/dry climate. This part should be focused on the most significant climatic proxies, unnecessary information should be moved into discussion or supplement. In the entire "Results" chapter results are mixed with interpretations (e.g., Lines 211-223). Reconsideration of used climatic proxies will also help to decrease the number of fossil data shown in Fig. 2, which looks too detailed. It should be also clear for the reader, whether discussed results are plotted in Fig. 2 or just taken from previously published studies (e.g., information on diatoms, L. 160). XRF PCA1 in Fig. 2 is not explained in the text. I am wondering if this proxy is important for the main outcome of the study. I would also suggest including the brief description of the age model in the "Results" section, not in the "Discussion".
- 3) better structure the "Discussion" part. It should be divided into two subsections, in accordance with the title of the manuscript (high-latitude events and seasonal trends). Shortening and better focusing is also needed.
- 4) improve figures:
 - a) Fig. 1. It is confusing that "Korallgrottan" is on the map (although the site is not mentioned in

the main text), but locations of Belgium speleothems, also used for the age model construction, are not shown. Different color labeling for ice cores, marine records, pollen sites and speleothems on the map could be useful. With the white area MIS 6 maximum extent of the ice-sheet is shown (not MIS 6 maximum extent).

b) Fig. 2 (see above)

c) Fig. 3. Supporting paleoclimatic information from aquatic proxies (from sedimentology, diatoms, algae) as well as statistical significance in Fig. 3A-B appear to be too detailed.

Other comments:

L. 38-40: abrupt events are linked to Greenland ice-sheet instabilities, but in the text both GIS and LIS are mentioned.

L. 50. Here "130-115 ka" is used to frame MIS 5e, but in the abstract "130,000 to 115,000 years ago".

L. 103. Explain what is meant with "cryosphere feedbacks". It seems that further in the text (L. 324) it is used as a synonym to "sea ice".

L. 111. "regression tree" (Fig. S3)?

L. 117. rephrase "Pollen of *Juniperus* shows values"

L. 121. Why *Picea abies* is used in the text, but *Picea* is in the Fig. 2?

L. 125-139. Results mixed with interpretations. It is difficult to understand, whether the discussed proxy data could be found in Fig. 2 or in Pliik et al. (L. 136, "aerophilic and halophilic diatoms"). Why enhanced catchment erosion is important? It is also not clear, why is it important that cooling "started just prior to the Tunturi event".

L. 140. comma after 17.98 m

L.140. The onset of Zone V "represents" or Zone V starts

L. 157. The event is even more

L. 157-159. "...even more distinct than the Tunturi event in the aquatic proxies..." which proxies are meant here? Statement is unclear and seems to belong rather to "discussion" part.

L. 163-167. The word "reconstruction" is used 5 times.

L. 173-174. "recovery of pine forest" Is this just one sample with increased presence of *Pinus* pollen? If yes, then is this result significant? Considering "enhanced noise and poor between-method robustness" (L. 192-193) for this time interval, I would be cautious interpreting the intra-event climatic variability. On the other hand, a gradual warming across the cold event is seen in the pollen data and in the T_{jul} reconstructions.

L. 186. situated

L. 182-193. Could be rephrased to emphasize the actual robustness of the temperature reconstructions.

L. 202. best analogues are found (remove "though moderate")

L. 211-226. Too detailed. Could be simplified and included into the discussion part.

L. 212. While "planktonic" is used here, "planktic" is used in L. 249, L. 710 and Fig. 4E.

L. 223-226. The statement is unclear. "T_{jan} should be interpreted with caution" or "shift towards warmer winters is a salient feature"?

L. 230-237. I find the new results from the Coral cave in Sweden crucial for the new timescale, therefore I would suggest mentioning this in the main text. Otherwise, the age model from Helmens et al. (2015) appears more plausible to me, considering the inferred late establishment of warm conditions in the eastern Nordic Seas and continuing deglaciation of the European ice sheets until mid-Eemian (e.g., Van Nieuwenhove et al., 2011).

L. 239-263. Generally, too detailed. Unnecessary descriptions (e.g., spatial mapping of d18O anomalies, AICC2012 chronology) should be removed.

L. 239. shows

L. 240. freshwater events

L. 241. SSTs at "the Eirik Drift" bring unnecessary details if you want to focus on Fig. 4 in this paragraph.

L. 248. based on negative anomalies or low d18O values

L. 254. Reference to Fig. 4E here is confusing. That the arrow indicates the LIS meltwater event in the d18O record from the East Greenland margin core is too superficial (age uncertainties are not

acknowledged).

L. 255. "lengthened the cooling" I believe that changes in AMOC/NADW (Fig. 4D) rather than SST fluctuations in the Labrador Seas, are more plausible in this paragraph.

L. 259-260. "effects of warming and hydrological cycle on top ocean buoyance" needs rephrasing

L. 260-261. What is meant by "continued Late Eemian events" seen in Fig. 4D?

L. 265-266. I suggest to change the wording of "the record on similar abrupt climate events is equivocal" to "records do not reveal a comparable sequence of abrupt climate events". This sentence should be combined with the next one.

L. 273-275. These lines about modeling should be combined with the paragraph below (L. 294-304). Could L. 265-275 and L. 294-304 be shortened and combined together?

L. 277-278. The meaning of the sentence is unclear. Does it mean "In contrast to the mid-latitude records, data from the high-latitude Sokli site reveal abrupt events within the Eemian"?

L. 308. Fig. 4G

L. 309-314. Interesting, but seems not relevant for the current study.

L. 316-317. Fig. 4 is referenced too often in these lines.

L. 329. suggest

L. 437. calcium

L. 439-441. The meaning of the sentence is unclear. Rephrase.

L. 690. Telford and Birks 2011

L. 706. Black arrows

Reviewer 1: *Salonen et al reconstruct Eemian climate from the important Sokli location in Northern Finland. The manuscript is well written.*

The summer temperature reconstruction seems reasonable, and events in the reconstruction can be related to climate events in from other proxy records. Although the authors have done a good job to validate their ability to reconstruct winter temperature, certainly one of the best I have seen, I remain somewhat sceptical about the winter temperature reconstruction. How much, for example, could changes in precipitation drive the reconstruction?

In modern ecology, winter temperature is established as a major control on the eastern limits of many deciduous trees in Europe (Sykes et al. 1996, and references therein). We have shown before (Salonen et al. 2012, 2014) and here (Supplementary Methods) that pollen–climate models are successful in identifying these winter indicators. By contrast, the effect of precipitation is less well established. Based on CCA tests, requested by Reviewer 1 (see p. 4), the independent effect of water balance is considerably smaller than that of winter temperature.

We note, however, that in the Fennoscandian climatic setting, it is likely that winter temperature and precipitation have generally co-varied in the past: both increase with the incursion of Atlantic low-pressure systems. While we consider winter temperature the stronger variable to reconstruct, this connection should make the variable selection less critical.

We stress that this first-order warming trend is the sole feature of winter temperature reconstruction we claim as significant, or use in our argumentation. In assessing the reconstruction, we place emphasis on the coherent response of the well-established oceanic taxa during the late Eemian, as well as the support from diatoms indicating shorter ice duration. As noted, the reconstruction is not likely to be compromised by precipitation variations, nor is it likely to be a spurious reflection (*sensu* Juggins 2013) of the negative trend in summer temperature, as the calibration-data correlation is low and has the opposite sign.

We have added a note in the Supplementary Methods about the smaller effect of water balance compared to winter temperature in partial CCA, as well as the likely tendency for the two variables to covary over time.

Reviewer 1: *Should (by analogy with the Holocene) the reconstructions be corrected for land uplift? Is the Eemian climate in northern Finland expected to have good climatic analogues in the modern climate? How might this affect the reconstructions, particularly of winter temperature?*

The history of Saalian deglaciation and Eemian land uplift in Finland is essentially unknown. However, drawing on the Holocene analogue, our study site lies in a region of moderate land uplift by Fennoscandian standards, at ca. 3–4 mm/a today. This rate is similar to southern Finland, where the Holocene uplift is well studied. Here, the total uplift during the Holocene has been ~100 m, and only ~30 m during the past 8 ka (e.g. Eronen and Ristaniemi 1992). Thus we estimate land uplift to have a negligible effect on our climate reconstructions.

With modern climate analogues, we did not have any strong expectation on how good they would be, as the Eemian climate of Northern Europe has not been studied in this detail before. However, based on the analogue distance metric used here (Fig. 3C,D), while the Eemian climate was very different compared to present at Sokli, for most of the sequence good modern analogues are found in warmer and more oceanic parts of Fennoscandia. Crucially, with regard to the reviewer’s question, this includes the steady T_{jan} rise of the mid-to-late Eemian.

Reviewer 1: *The manuscript should state where the data and code needed to reproduce the results are to be archived.*

For our revised manuscript, we have prepared two Supplementary Data files – an Excel file with the pollen and macrofossil data, model calibration data, and the climate reconstructions, and an R code file for running the climate reconstructions. These data will be uploaded on figshare upon publication of the paper.

We have added a Data Availability statement in our revised manuscript. We will update this with the exact figshare links for the final draft.

Reviewer 1: *Perhaps this is a bizarre journal policy, but separating the figures from their caption and having both remote from the text that refers to them is a certain way to frustrate and annoy the reviewer. Please do not do this.*

We apologize for having frustrated the reviewer. In our revised manuscript, we have placed the figures, with their captions, in the appropriate places within the text.

Reviewer 1: *182 The modern pollen analogue distances (Fig. 3C) for fossil pollen samples are small -> short*

Done.

Reviewer 1: *224-226 This sentence is contradictory.*

The point we try to make here is that the first-order trend of the winter reconstruction (rising T_{jan} across the Eemian) is seen as significant, however we otherwise urge caution in interpreting the T_{jan} curve – due to the methodological challenges discussed. Our worry is that a less methodologically oriented reader might try to overanalyze small features of the T_{jan} reconstruction, hence this statement is important.

We have rephrased this sentence to be clearer.

Reviewer 1: *297 global climate heading towards Eemian Interglacial levels - Climate does not have levels, temperature instead?*

Replaced “climate” → “temperatures”.

Reviewer 1: *324 This would appear to contradict the rest of the paragraph. At 67N, one might expect that Arctic Sea ice would be important, and thus the indirect influence of summer insolation affect winter temperatures. But the reconstruction shows rising winter temperatures following winter insolation. Maybe "are pivotal" should read "should be pivotal"*

Rephrased as “should be pivotal”, as suggested. The intended meaning is that based on available transient modelling, either falling or rising winter temperature trend appears possible, critically depending on how far south the effect of sea-ice feedbacks reaches.

Reviewer 1: 412 testing the correlation and independent effect of the reconstructed variables in the calibration data - unclear. Is this with an ordination?

This refers to partial CCA (pCCA) permutation tests, as described in Supplementary Methods. However we have removed this test as unnecessary, as noted in a reply below (p. 4).

Reviewer 1: 709 *Neogloboquadrina pachyderma* -> is this sinistral or dextral (or both)

These are sinistral *N. pachyderma*, however the authors of the data shown here (Zhuravleva et al. 2017) call these simply *N. pachyderma*, citing the latest taxonomic status where the sinistral and dextral forms are considered separate species.

Reviewer 1: Supplementary material

The significance of the partial CCA is of little importance, especially as autocorrelation in the calibration set has not been accounted for. The proportion of variance explained by each variable and their covariance would be more useful guide to the importance of each seasons temperature.

What is the ratio of the first to second eigenvalue for ordinations constrained by just summer or winter temperature (see Juggins 2013)

The reviewer refers to the guideline of Juggins (2013) that the ratio between the eigenvalue of the axis constrained by the reconstructed variable and the eigenvalue of the first unconstrained axis (λ_1/λ_2) should be > 1 . It has been noted earlier (Salonen et al. 2014) that the λ_1/λ_2 values tend to be low in Northern European pollen–climate datasets. This extends to summer temperature – the classical and uncontroversial reconstructed variable – for which Salonen et al. (2014) report a λ_1/λ_2 of 0.46. We find a similar value for T_{jul} (0.49), although in our data λ_1/λ_2 for T_{jan} is higher (0.56). The reasons for the low λ_1/λ_2 values are uncertain. One possibility is that the large and complex pollen datasets are not ideal for CCA, a method underpinned by unimodal modelling. This is hinted at by our cross-validation results where non-parametric methods (RF, BRT) have a better performance compared to classical, unimodal methods (WA, WA-PLS, MLRC).

As suggested by Reviewer 1, we tested the variance explained by T_{jul} , T_{jan} and water balance (WAB) in CCA, and variance explained independently of covariables in pCCA. The variance explained is 6.0% for T_{jul} , 6.7% for T_{jan} , and 5.8 % for WAB, while the variance explained independently (of the other two climate variables) is 2.0% for T_{jul} , 4.8% for T_{jan} , and 1.9 % for WAB. Thus the independent effect of T_{jan} is *greater* than that of T_{jul} , and also greater than that of WAB. This result may be optimistic, as our calibration data extend into Central–Western Europe where winter temperature is the dominant variable, while our reconstructions are from a boreal environment where summer temperature is dominant. Thus, as noted earlier (p. 2), we have placed great emphasis on the response of specific winter temperature indicator taxa within boreal environments, as well as supporting evidence from diatoms.

In response to the reviewer’s comment, we have removed the pCCA permutation tests from Methods. They are perhaps redundant here, as we proceed test our models’ predictive ability more rigorously, using cross-validations with the calibration data. Also, as noted above (p. 2) we show the pCCA results to justify the choice of T_{jan} over WAB.

Reviewer 2: *The Last interglacial (LIG, ~129-116 ka) offers an opportunity to study the response of the Greenland and Antarctic ice sheets to a warmer-than-preindustrial high-latitude climate. While climate changes above the polar ice sheets and at the surface of the oceans across the LIG have been increasingly documented over the past years, there is still a lack of high-resolution paleodata informing on climate changes in land-areas, especially in the high latitude regions. This hampers our ability to fully understand the climate feedbacks as well as the processes associated to land-sea contrast that occur in those regions. The high latitudes are particularly sensitive to changes in the radiative forcing and act as amplifiers of climate change. Studies presenting new insights on high-latitude terrestrial climate changes across the LIG are thus needed.*

In this context, the study of J. Sakari Salonen and co-authors is very relevant and will be of great interest for the paleoclimate community since it presents seasonal temperature reconstructions from the Sokli lacustrine sediment record from Northern Finland using pollen-climate calibration models. They emphasize that their climate reconstructions (January and July temperatures) suggests 1) the existence of two abrupt cooling events in this region and 2) decoupled trends between the summer temperature evolution and the winter temperature evolution across the LIG.

Overall the authors have developed an interesting study in a well-written paper accompanied with sound explanations of the methodology and appropriate figures. However I have a doubt that enough evidences are provided for its conclusions and that the claims are novel enough to guaranty its publications in Nature Communications. Indeed, a large part of their manuscript is dedicated to the description and discussion of the two abrupt cooling events referred to as the Värriö and Tunturi events. As interesting as the identification of these events is, they have already been described in the Sokli record (based on other proxies) in previous publications (e.g. Helmens et al. 2015, Plikk et al. 2016). Actually as far as I understand the authors refer a couple of time to the Tunturi event identified previously in the introduction in several places without naming it (line 84 and line 97).

Also, in several places in the manuscript (further described below), the proposed statements are “overselling” what can actually be seen from their reconstructions. For instance it is stated in the the abstract that “the peak interglacial conditions are intersected by two strong cooling and drying events”. However (1) according to the dating of these events, we cannot rule out that those events are not happening during purely interglacial conditions e.g. The Tunturi event could be associated to the penultimate deglaciation. Also to me (2) the signature of the Värriö event in the pollen markers is neither strong nor clear and as a result neither it is in the January and July temperature reconstructions.

Thus, publishing those results in a more specialized journal might be appropriate. I detail below some important comments that the authors should consider when preparing a revised version of their manuscript.

Compared to Helmens et al. (2015), this paper has a different focus, and the data are largely new. Helmens et al. (2015) was a fast-tracked publication using a preliminary, lower-resolution version of some of these data. It focused solely on the Early/Mid-Eemian Tunturi event.

Plikk et al. (2016) is a palaeolimnological paper and not proximally a climate reconstruction study. We refer to it occasionally to validate our pollen-based palaeoclimate reconstructions, especially where the diatom data provide helpful insights on ice cover length and lake levels.

By contrast, this paper describes the climate of the entire Eemian, including both long and short term variation, and the seasonality of climate. For this purpose, we have developed a significantly improved palaeoclimate dataset:

- We have doubled the resolution over the Mid-to-Late Eemian compared to Helmens et al. (2015), with new proxy analyses. With this we aim to resolve any further abrupt events (beyond the Tunturi event) and to rule out similar shifts over the rest of the sequence.
- We use a considerably more advanced climate reconstruction method, with a multi-method ensemble approach (only WA in Helmens et al. 2015).
- We reconstruct both summer and winter temperature (only summer in Helmens et al. 2015). Assessing the significance of the winter climate is a key component in this paper.
- We have a significantly changed chronology, based on speleothem data not available when Helmens et al. (2015) was written: Moseley et al. (2015), Vansteenberghe et al. (2016, 2017), and the Korallgrottan speleothem (Holzkämper et al., *in prep.*).

Using these new data, we place major focus on long-term climate trends and their implications – entirely outside the scope of Helmens et al. (2015) – and the closing half of our Discussion deals with these. Here, multiple important insights are presented:

- We present a benchmark series of Eemian climate from the northern high latitudes, showing a distinct, mixed influence of short and long term variations in the North Atlantic circulation in conjunction with secular insolation forcing. This is the first study to document this climate response, due to the lack of earlier land-based high-latitude data.
- We suggest that the earlier Eemian climate data from the European continent, clustered in the mid latitudes, are not representative for a major northern portion of Europe.
- We also find significant deviations from Eemian climate modelling, with implications to validation of the models, and especially their incorporation of cryosphere feedbacks.
- We present new understanding about the climatic evolution leading to the last glacial inception in Europe. Vitaly, these are the first such data from land areas proximal to the glaciation.

In editing our revised manuscript, we have shifted the focus towards these new findings, with many sections dealing with the abrupt events shortened. The new evidence about the conditions leading to last glacial inception are now included in the Abstract.

However, we were compelled to also revisit the abrupt events, since we had a much improved climate reconstruction (higher proxy resolution, improved reconstruction methods and chronology). This was especially needed as new marine data published after Helmens et al. (2015) provided a more coherent correlation between Sokli, the North Atlantic, and specific Greenland meltwater events. Also, we were now able to confidently pick up the Värriö event, which crucially depended on the higher data resolution used here vs. Helmens et al. (2015).

To summarize, the underlying fossil data are partly the same as in Helmens et al. (2015), but now analyzed at a fundamentally higher resolution, while the climate reconstructions are entirely new, as is the chronology. The focus of the paper and the major findings are new.

Reviewer 2: MAJOR COMMENTS:

- *Abrupt cooling events in the Sokli record*
- *In this manuscript, the authors report on two cooling events that they evidence at the*

beginning (the Tunturi event) and at the end of the LIG (the Värriö event) in the pollen records. Both events have been previously described in studies based on other proxies (e.g. Helmens et al. 2015, Plikk et al. 2016). Based on the fossil data presented in Figure 2, I find that the imprint of the Värriö event in those dataset is not straightforward. Therefore I am uncomfortable with their statement in line 154 “...shifts are recorded in all proxies”. The authors need to be more specific here about which proxies they are talking about.

I have a similar criticism regarding this event looking now at the seasonal temperature reconstructions. I do not think that the level of variability observed in the records allows to identify with confidence the Värriö cooling event both the median T_{jan} and T_{jul} reconstructions. Indeed the authors claim that the abrupt onset represent the largest T_{jul} fall in all reconstructions during the prolonged warm interval following the Tunturi event but how significant is this fall? On how many data point is this based?

The authors need to build a stronger a more convincing case. In particular it would be helpful to ease the reading of the results to have additional figures that display all the temperature reconstructions based on the different methods in such a way that is easy and straightforward to look at them one by one with both the raw and smoothed reconstructed temperature data.

The fossil imprint of the Värriö event consists of a sharp minimum in total arboreal pollen and peaks in *Betula nana* and *Sphagnum* at the sharp onset of the event. The arboreal pollen minimum across two samples (1737–1739 cm) is the most important feature, with the arboreal/non-arboreal pollen ratio falling to a value (2.57) below the lowest value during the Tunturi event (4.85). The event is also strongly seen in green microalgae (this study), with maxima in shallow-water *Spirogyra* and cold-oligogrophic *Pediastrum integrum*, and a corresponding minimum in other green algae, as well as in diatoms (Plikk et al. 2016). The event is much longer in the aquatic proxies, and this is the basis of the event length shown with the pink bar.

In the summer reconstruction (Fig 3A) the event is distinct. We emphasize that the reconstructions in Fig. 3A are unsmoothed raw data, with a high resolution. Following the recovery from the Tunturi Event and until the end-Eemian fall in temperature (between 16–22 m depth), the Värriö Event is the only fall of the ensemble median to present levels. Otherwise the median stays considerably above present, without exception across ca. 140 samples. Within the half-metre leading up to the event median T_{jul} varies within 14.6–15.5°C across 12 samples (mean = 14.9°C). During the sharp onset of the Värriö event, T_{jul} falls to 12.9 and 13.2°C in two consecutive samples. During the half-metre following this T_{jul} dip, temperature is again higher and stable at 13.8–15.3°C across 19 samples (mean = 14.5°C). Thus the event sees a T_{jul} fall of 1.5 to 2.0°C, beyond the range of variation before or after the event.

The winter reconstruction (Fig. 3B) is likely to largely reflect major changes in winter temperature indicators across the Eemian (see replies to Reviewer 1). We would not attempt to determine the presence or absence of abrupt events – with an unspecified seasonality – based on the T_{jan} reconstruction.

We have done the following changes to the manuscript:

- We describe more clearly the fossil record of the Värriö event, as outlined above.

- We describe the absolute magnitude of the Värriö event in the T_{jul} reconstruction as was already done for the Tunturi event: as the amplitude in the ensemble median, and as the range of amplitudes in the individual methods.
- We have increased the size of Fig. 3 by ca. 50 %, to ease the assessment of the reconstructions. We are open to adding separate panels for the abrupt events, however we are uncertain that this is needed with the entire figure enlarged, with the improved description of the anomalies in the text, and with the raw data in a supplement.

Reviewer 2: *Regarding the Tunturi event, the authors claim also that all methods show distinct cold peaks within the Tunturi events. Again, the author need to build a stronger case if they want to be so definite that the variations that are reconstructed within the cooling event represent significant climate signals and are not simply related to the noisiness of the signal. Otherwise, the authors should toned down their statement and point out clearly the limitations of their observations.*

The evidence based on which we suggest the two-pronged structure are:

- A spike of *Pinus* pollen and a *Pinus* stoma halfway into the event, with a corresponding *Betula* pollen minimum (Fig. 2). This represents a short-lived reversal of the most major shift (*Betula* increase at the expense of *Pinus*) occurring at the onset of the Tunturi event.
- The median of all T_{jul} reconstructions appears to show two sharp cooling maxima, at the onset of the event and a second one about two-thirds into the event (Fig. 3A).
- A warming and rise in lake levels during central part of the event is also suggested by the diatoms (Pliikk et al. 2016).

Given these multiproxy evidence, we think the two-pronged structure is there, but we allow that this might be open to different interpretations.

In our revised manuscript, we present more clearly the above evidence based on which we suggest the two cooling peaks. However, we use more cautious wordings in suggesting this (“appears to have”, “possible”).

Reviewer 2: *I believe that because of the limitations of the temperature reconstructions raised above, the authors should toned down a few of their statements throughout the manuscript, for instance:*

Line 37: “...the peak interglacial conditions are intersected by two cooling events” (note also as discussed also in the manuscript that the early cooling event may not represent purely interglacial conditions but be tied to the preceding termination. The authors should thus not use the expression “peak interglacial”).

Line 101: “Our climate reconstruction provide a robust recorded of repeated Eemian abrupt events...”.

We have rephrased L37 to say simply “interglacial conditions”. We have also edited analogous statements throughout the manuscript, to say that the abrupt events intersect “interglacial” or “warm interglacial” conditions instead of “peak interglacial” conditions.

Reviewer 2: *Age model of the Sokli record*

It is essential that more information is provided regarding the age model of the record. In particular I find it very problematic that the speleothem records used to define the two tie points at the onset and at the end of the LIG in the Sokli record are not shown in any of

the figures. The resulting alignment of the pollen record onto the speleothem records need to be showed. The climatic assumption used to use the dates of onset and end of the LIG as seen in the Northern Alps and Belgium speleothem needs to be clearly spelt out and also discussed. In particular the authors should provide the limitations associated with their age model and also try and quantify the associated uncertainties.

We have significantly expanded the presentation of the chronology its uncertainties:

- We show all the stalagmite data which our chronology relies on in a new panel in Fig. 4.
- We have added all the used stalagmite sites on the map (Fig. 1).
- We describe better the underlying hypotheses in using the stalagmite stable isotope events as chronological tie points (Methods text).
- We have prepared an age-depth model which allows the estimation of sample-specific age errors. This age model accounts for the estimated uncertainties in the tie point alignments, as well as additional uncertainties introduced by variation in deposition rate.

While noting these uncertainties, we wish to stress that we have been careful to avoid any argumentation that critically depends on absolute age estimates. Correlation of the Tunturi event with Atlantic proxies is done based on the sequence of events, not the absolute timing (which however does coincide, within the estimated age errors). Correlation of the Värriö event with marine events is not attempted. Apart from the cold events, we deal with the first-order trends in seasonal climates across the entire Eemian, how these differ from other records and modelling, and the significance of these differences.

Our revised manuscript slightly alters the chronology, as we formulated the underlying hypotheses. In our original submission, the Moseley et al. (2015) and Korallgrottan speleothems were together used to argue for an onset of the Eemian at ~130 ka. We now tie the onset directly to the Moseley et al. (2015) speleothems, and their exact dating of 130.9 ka (i.e., change of 0.9 ka), while the Korallgrottan dating is used to argue that Sokli was deglaciated by this time.

Reviewer 2: *Global temperature during the LIG*

Lines 32, 51, 53, 61: The authors state that the LIG was globally warmer than today. The authors need to be very careful here and rephrase their statements. I believe that they are based on the estimate of ~1.5°C higher than today provided by the Turney and Jones (2010) climate synthesis. However, this estimate is attached to very important limitations leading to a quantitative estimate of the global LIG warmth that is unreliable and possibly overestimating the global LIG warming. Indeed, this estimate is based on the assumption that the LIG peak warmth is synchronous across the world while there are multiple evidences now in paleo records that this is not correct (e.g. Govin et al. 2012, Capron et al. 2014, Hoffman et al. 2017). As a result the Turney and Jones (2010) synthesis does not represent a climate state at any specific time periods across the LIG but a virtual (and unrealistic) image on the LIG peak warmth across the globe. The latest global SST synthesis based on coherent chronologies and allowing for asynchronous climate changes across the LIG suggest a global SST average that is indistinguishable from the 1995-2014 mean (Hoffman et al. 2017; the authors should also consider adding a reference to this paper).

However, it is safe to state (and very relevant to their study) that the high latitudes were warmer than today during the LIG as seen in polar records (e.g. Masson-Delmotte et al. 2011; Landais et al. 2016; NEEM community members 2012), as well as in the latest syntheses of marine records from the North Atlantic and Southern Ocean (Hoffman et al.

2017; Capron et al. 2014, 2017) that provide a spatio-temporal evolution of the LIG climate and are based on coherent chronologies.

Thus I suggest the authors to rephrase their statements (and possibly refer to high-latitude temperature estimates instead of global temperature).

We thank the reviewer for pointing out the Hoffman et al. (2017) study. Based on a chronologically coherent dataset of sea-surface temperature proxies, Hoffman et al. (2017) suggest a smaller global anomaly of $0.5 \pm 0.3^\circ\text{C}$ compared to pre-industrial, while stressing that contrary to most climate modelling, based on the data there was indeed a global anomaly.

We have done the following changes to the introductory paragraph:

- We summarize the global pattern as “temperatures above preindustrial levels by an estimated 0.5°C globally and by up to $4\text{--}5^\circ\text{C}$ in the Arctic”, with reference added to Hoffman et al. (2017).
- In the first sentence, “globally warmer than present climate” → “widespread climatic warming”
- Later in the paragraph, “remarkably similar” → “similar”, when drawing parallels between the Eemian and late 21st century.
- Removed the last sentence of the paragraph, referring to a “prominent case of global warming”.

Reviewer 2: MINOR COMMENTS:

Title, lines 31 & 50: The “Eemian Interglacial period” is an awkward formulation. The authors should refer to either “the Eemian” or “the last interglacial period” (or simply “the Last Interglacial”) but not a combination of both.

In our revised manuscript, we use “the Eemian Interglacial period”, in full for clarity, when the concept is first introduced (Title, start of Abstract, start of Introduction), while in later instances we use simply “the Eemian”.

Reviewer 2: Line 31: *I appreciate that depending on the definition for the LIG used, the time interval can vary (as shown in Figure 1 from Govin et al. 2015). But I suggest the authors to use here the time interval “129-116 ka” given in the last IPCC report by Masson-Delmotte et al. (2013) defined based on eustatic sea level variations.*

Done.

Reviewer 2: Line 54: *“significantly reduced ice sheet in Greenland”; this statement needs to be hampered and the authors should cite Dutton et al. (2015) instead of Dutton and Lambeck (2012). They provide an updated assessment and summary of the LIG sea level changes and the likely respective contributions of the Greenland and Antarctic ice sheets to the sea level rise. Together with the results from the NEEM community members (2012), it seems that the contribution of Greenland to the LIG global sea level rise was more modest than previously stated older studies.*

We are not sure the evidence summarized in Dutton et al. (2015) requires a change in our statement. Their estimated sea-level highstand (6–9 m) is the same we had summarized based on Kopp et al. (2009) and Dutton and Lambeck (2012). Likewise they also suggest a

“substantial (but not complete) retreat of the southern sector [of the GIS]”, while Antarctic contribution is suggested by the sea-level signal but not currently backed by geological evidence. This was precisely the reasoning behind the statement in our Introduction, i.e., substantial ice-sheet reduction in Greenland and possibly in Antarctica.

We have, however, replaced Dutton et al. (2015) as the citation for the sea-level estimate, as they provide an up-to-date review of the available evidence.

Reviewer 2: *Line 61: follow my previous comment, the authors should write: “...to HIGH-LATITUDE climate warming...”*

Simply “climate warming” is perhaps fair, considering the global anomaly suggested by Hoffman et al. (2017), while smaller than suggested by some earlier studies.

Reviewer 2: *Line 74: Northern Hemisphere instead of Northern-Hemisphere.*

Done.

Reviewer 2: *Line 97: “multi-proxies” is very vague, the authors should be more specific here.*

Elaborated with a list of proxies.

Reviewer 2: *Line 185: “...that the coring site was situations in the central part...” this sentence needs to be revised.*

Done.

Reviewer 2: *Line 263: I agree that chronological uncertainties are too large to draw confident links between the Värriö event and changes in marine records at that time, but it would be worth considering to cite also to the Mokeddem et al. (2014) paper which suggests repeated cold water-mass expansions into the subpolar latitude throughout the LIG.*

Done. We have also added a citation to Mokeddem et al. (2014) at the end of Discussion, when discussing the climatic conditions leading to last glacial inception.

Reviewer 2: *Line 678: Caption of Figure 2. Please be more specific about the colored bars and which events they indicated e.g. Yellow bar indicates the Tunturi Event and pink bar indicates the Värriö event.*

Done.

Reviewer 2: *Line 704: What are the LOESS smoothers? This needs to be clarified.*

LOESS is a locally weighted scatterplot smoother. The smoothers are specified with two parameters, span (here, 0.03) and the number of added, robustifying iterations (here, 1).

We have added the used LOESS parameters in the caption to Fig. 4.

Reviewer 2: Line 713: Change sentence into “NEEM and NGRIP ice d18O records (38, 39) displayed on the the AICC2012 chronology (11, 44, 45).

Done.

Reviewer 2: Line 718: “the time scales used are floating chronologies” What do you mean by floating?

These fossil sequences do not have absolutely radiometric dating of their own, however estimates exist on the absolute chronological *durations* of the pollen zones. Thus the sites have absolute internal chronologies while leaving open where these internal chronologies should be anchored relative to present – hence the chronologies “float”.

Reviewer 3: *Saloranta and co-authors present quantitative reconstructions of July and January mean air temperature during the Eemian interglacial, based on high-resolution pollen record from the Sokli site in northern Finland. The work is based on previously published findings (Helmens et al., 2015; Plikk et al., 2016; Kylander et al., 2018) and the new data from the study site. The temperature reconstructions are statistically significant, derived from a number of pollen-climate calibration models and extensively validated, e.g., by testing statistical significance. The combined results presented in the manuscript (e.g., complete temperature dataset for July and January) could contribute to the understanding of interglacial climates as well as used in models. Of particular interest are the inferred decoupled trends in summer and winter temperature across the Eemian, ascribed to combined influence of insolation and oceanic forcing. Due to general paucity of available high-latitude terrestrial records and the particularly high-resolution of the Sokli sequence, the reconstructed winter temperature record is unique and could influence thinking in the field. The study also investigates the succession of climatic events on land, linking them to documented shifts in the oceanic circulation.*

The paper is generally well-written, but in some aspects is too detailed. Results and interpretations appear sound, although sometimes they are mixed together. Also, the introduction should be more specific, the storyline in the discussion more focused. Some figure corrections are also essential. My suggestions to improve the manuscript are: 1) shorten the introduction, making it more specific. Authors should clearly state the problem they want to investigate and remove the unnecessary descriptions.

Following the suggestions of the reviewer, we have done these changes to the Introduction:

- We have shortened the mid section of the Introduction, describing the existing proxy record of the Eemian abrupt events, as these largely repeated parts of Discussion. This also helps shift the focus towards the long-term climate trends, an intended focus and an entirely new contribution in this paper.
- At the end of Introduction, we state the goal of this paper (i.e., study of high-latitude Eemian climate including both long-term seasonal trends and the record of abrupt events).

Reviewer 3: 2) shorten the "Results". I found this chapter to be particularly dense

reading, with the description of all the proxies (even those that are not used for temperature reconstructions). This part also seems to be somewhat repetitive, considering the earlier studies from the Sokli site. In the subchapter "Fossil record and vegetation changes" temperature-related proxies are mixed with sedimentological data and descriptions about wet/dry climate. This part should be focused on the most significant climatic proxies, unnecessary information should be moved into discussion or supplement. In the entire "Results" chapter results are mixed with interpretations (e.g., Lines 211-223). Reconsideration of used climatic proxies will also help to decrease the number of fossil data shown in Fig. 2, which looks too detailed. It should be also clear for the reader, whether discussed results are plotted in Fig. 2 or just taken from previously published studies (e.g., information on diatoms, L. 160).

XRF PCA1 in Fig. 2 is not explained in the text. I am wondering if this proxy is important for the main outcome of the study. I would also suggest including the brief description of the age model in the "Results" section, not in the "Discussion".

We broadly agree with these suggestions. We have done significant edits to the Results to simplify and shorten their presentation:

- Some non-essential palaeoecological and sedimentological detail has been removed.
- The previous allowed numerous taxa to be removed from Fig. 2.
- XRF PCA1 is also removed from Fig. 2.
- With Fig. 2 now smaller, we rotated it by 90° to landscape orientation.
- We have sharpened the division between Results and Discussion. We have (A) moved the age-model paragraph from Discussion to Results, and (B) moved the interpretation and validation of our reconstructions to the start of Discussion. The Results now have no references to previous studies on this sequence.

Reviewer 3: *3) better structure the "Discussion" part. It should be divided into two subsections, in accordance with the title of the manuscript (high-latitude events and seasonal trends). Shortening and better focusing is also needed.*

The Discussion would indeed benefit of subsections, but they are unfortunately forbidden in the Nature Communications author instructions. Instead, we have tried to make the Discussion have a clear transition from the first half (abrupt events) to the second half (long-term trends). We mark this transition by starting a paragraph with “In long-term Eemian climate trends...”.

We have significantly shortened parts of Discussion. This is especially done at the discussion of the abrupt events, to shift the focus towards the long-term trends.

Reviewer 3: *4) improve figures:*

a) Fig. 1. It is confusing that "Korallgrottan" is on the map (although the site is not mentioned in the main text), but locations of Belgium speleothems, also used for the age model construction, are not shown. Different color labeling for ice cores, marine records, pollen sites and speleothems on the map could be useful. With the white area MIS 6 maximum extent of the ice-sheet is shown (not MIS 6 maximum extent).

b) Fig. 2 (see above)

c) Fig. 3. Supporting paleoclimatic information from aquatic proxies (from sedimentology, diatoms, algae) as well as statistical significance in Fig. 3A-B appear to

be too detailed.

We agree with these suggestions. We have done the following changes:

- Fig. 1: The sites shown on the map have been updated. With the addition of the speleothem panel in Fig. 4, we now show all the speleothem sites on the map. As suggested, we are using different colours for different proxy types. The MIS 6 ice sheet is now marked as “MIS 6 max. glaciation” which is perhaps clearer.
- Fig. 2: Significantly simplified – see previous reply.
- Fig. 3: We have removed the supporting information and statistical significance bars. These intervals are described in the manuscript text, and this is perhaps sufficient.

Reviewer 3: *Other comments:*

L. 38-40: abrupt events are linked to Greenland ice-sheet instabilities, but in the text both GIS and LIS are mentioned.

We have rephrased this as simply “linked to disturbances in the North Atlantic circulation regime”. Various evidence point to both GIS and LIS causes of these disturbances, and we go into these in the Discussion.

Reviewer 3: *L. 50. Here "130-115 ka" is used to frame MIS 5e, but in the abstract "130,000 to 115,000 years ago".*

Following a suggestion from Reviewer 2, we now use the interval 129–116 ka. The use of “129,000–116,000 years ago” in the first instance is intentional, as the abbreviation “ka”, in our experience, tends to be unfamiliar to non-geologists.

Reviewer 3: *L. 103. Explain what is meant with "cryosphere feedbacks". It seems that further in the text (L. 324) it is used as a synonym to "sea ice".*

We have rephrased this as “ice-sheet dynamics”. With sea-ice feedbacks the news from this paper is really the *lack* of strong evidence (while in climate modelling sea-ice feedbacks dominate the winter climate evolution in the European Arctic).

Reviewer 3: *L. 111. "regression tree" (Fig. S3)?*

Done.

Reviewer 3: *L. 117. rephrase "Pollen of Juniperus shows values"*

Done.

Reviewer 3: *L. 121. Why Picea abies is used in the text, but Picea is in the Fig. 2?*

These apparent inconsistencies are due to whether we discuss forest species or identified microfossils. Technically speaking, the fossil pollen were identified as representing *Picea* at the genus level (hence “*Picea*” in Fig. 2). However when speaking of forest composition, in the

present palaeoecological context it is clear these pollen must have been produced by the Norway spruce (hence "*Picea abies*" in text).

Reviewer 3: L. 125-139. *Results mixed with interpretations. It is difficult to understand, whether the discussed proxy data could be found in Fig. 2 or in Pliikk et al. (L. 136, "aerophilic and halophilic diatoms"). Why enhanced catchment erosion is important? It is also not clear, why is it important that cooling "started just prior to the Tunturi event".*

This is resolved with the restructuring of Results and early Discussion (see reply on p. 13). We now first present our own Results, and only refer to further evidence from Pliikk et al. (2016) in Discussion. We have removed the mention of catchment erosion.

We have kept the the mention of the "cooling prior to the Tunturi event". The aquatic record shows a distinct response before the extremely abrupt response of the pollen record and the pollen-based reconstruction, and we regard this as significant complementary information. We have edited this part to make clearer how this information complements the pollen record.

Reviewer 3: L. 140. *comma after 17.98 m*

Done.

Reviewer 3: L.140. *The onset of Zone V "represents" or Zone V starts*

Rephrased as "The onset of...".

Reviewer 3: L. 157. *The event is even more*

Done.

Reviewer 3: L. 157-159. *"...even more distinct than the Tunturi event in the aquatic proxies..." which proxies are meant here? Statement is unclear and seems to belong rather to "discussion" part.*

We have moved this section to Discussion, as noted above. We have also clarified the wording.

Reviewer 3: L. 163-167. *The word "reconstruction" is used 5 times.*

Rephrased two instances as "interval" and "curves".

Reviewer 3: L. 173-174. *"recovery of pine forest" Is this just one sample with increased presence of Pinus pollen? If yes, then is this result significant? Considering "enhanced noise and poor between-method robustness" (L. 192-193) for this time interval, I would be cautious interpreting the intra-event climatic variability. On the other hand, a gradual warming across the cold event is seen in the pollen data and in the T jul reconstructions.*

See previous reply (p. 7); we now note more clearly the evidence for the mid-event recovery, but also use more cautious phrasing for this feature.

Reviewer 3: L. 186. *situated`*

Done.

Reviewer 3: *L. 182-193. Could be rephrased to emphasize the actual robustness of the temperature reconstructions.*

This is a good idea – we have rephrased this paragraph to clearly state that the short analogue distances suggest the reconstructions are generally robust.

Reviewer 3: *L. 202. best analogues are found (remove "though moderate")*

Done.

Reviewer 3: *L. 211-226. Too detailed. Could be simplified and included into the discussion part.*

As noted above, we have moved this to the Discussion. We have also simplified the description of the diatom data.

Reviewer 3: *L. 212. While "planktonic" is used here, "planktic" is used in L. 249, L. 710 and Fig. 4E.*

Rephrased as “planktic”.

Reviewer 3: *L. 223-226. The statement is unclear. "T jan should be interpreted with caution" or "shift towards warmer winters is a salient feature"?*

We have rephrased this statement (see earlier reply, p. 3).

Reviewer 3: *L. 230-237. I find the new results from the Coral cave in Sweden crucial for the new timescale, therefore I would suggest mentioning this in the main text. Otherwise, the age model from Helmens et al. (2015) appears more plausible to me, considering the inferred late establishment of warm conditions in the eastern Nordic Seas and continuing deglaciation of the European ice sheets until mid-Eemian (e.g., Van Nieuwenhove et al., 2011).*

See reply on chronology on p. 8. We now show the Coral Cave dating in Fig. 4 (together with other speleothem data) and discuss in greater detail the uncertainties and underlying assumptions of the age model.

Reviewer 3: *L. 239-263. Generally, too detailed. Unnecessary descriptions (e.g., spatial mapping of $d18O$ anomalies, AICC2012 chronology) should be removed.*

We agree – several minor details have been removed from this paragraph.

Reviewer 3: *L. 239. shows*

Done.

Reviewer 3: L. 240. *freshwater events*

Done.

Reviewer 3: L. 241. *SSTs at "the Eirik Drift" bring unnecessary details if you want to focus on Fig. 4 in this paragraph.*

Rephrased.

Reviewer 3: L. 248. *based on negative anomalies or low d18O values*

Fixed.

Reviewer 3: L. 254. *Reference to Fig. 4E here is confusing. That the arrow indicates the LIS meltwater event in the d18O record from the East Greenland margin core is too superficial (age uncertainties are not acknowledged).*

We agree the arrow indicating the suggested LIS event is confusing, and we have removed it from the figure. We now phrase more simply that a LIS outburst is suggested to have prolonged this event, with reference to Zhuravleva et al. (2017) who propose this based on alignment of their own records with the Eirik Drift sequence.

Reviewer 3: L. 255. *"lengthened the cooling" I believe that changes in AMOC/NADW (Fig. 4D) rather than SST fluctuations in the Labrador Seas, are more plausible in this paragraph.*

Rephrased "lengthened the cooling" → "prolonged the event".

Reviewer 3: L. 259-260. *"effects of warming and hydrological cycle on top ocean buoyance" needs rephrasing*

We rearranged this sentence to be clearer.

Reviewer 3: L. 260-261. *What is meant by "continued Late Eemian events" seen in Fig. 4D?*

This refers to the sharp spikes seen in the Eirik Drift proxies following the longer event at ca. 125 ka. We now give two specific ages (117 and 120 ka) showing these sharp anomalies.

Reviewer 3: L. 265-266. *I suggest to change the wording of "the record on similar abrupt climate events is equivocal" to "records do not reveal a comparable sequence of abrupt climate events". This sentence should be combined with the next one.*

Done.

Reviewer 3: L. 273-275. *These lines about modeling should be combined with the*

paragraph below (L. 294-304). Could L. 265-275 and L. 294-304 be shortened and combined together?

We have significantly shortened the sections pointed out by the reviewer.

Reviewer 3: *L. 277-278. The meaning of the sentence is unclear. Does it mean "In contrast to the mid-latitude records, data from the high-latitude Sokli site reveal abrupt events within the Eemian"?*

This is indeed what is meant. We have rephrased this sentence to be more straightforward.

Reviewer 3: *L. 308. Fig. 4G*

Fixed.

Reviewer 3: *L. 309-314. Interesting, but seems not relevant for the current study.*

We are open to removing this passage, however we would argue it is relevant. These different interpretations imply an uncertainty of a few millennia to the onset of winter cooling in the mid latitudes, and comparing the winter trends is the focus of this paragraph. However, these uncertainties do not affect the conclusion that the winter temperature trends over the Late Eemian are opposite between Sokli and mid-latitude Europe.

Reviewer 3: *L. 316-317. Fig. 4 is referenced too often in these lines.*

Fixed.

Reviewer 3: *L. 329. suggest*

Fixed.

Reviewer 3: *L. 437. calcium*

Fixed.

Reviewer 3: *L. 439-441. The meaning of the sentence is unclear. Rephrase.*

We have rearranged this sentence – it should now be clearer how this follows from the preceding argumentation.

Reviewer 3: *L. 690. Telford and Birks 2011*

Fixed – this reference is removed from the caption as the significant intervals are only described in text.

Reviewer 3: *L. 706. Black arrows*

Fixed.

In addition to the changes described above, we have done some minor further edits to the manuscript and the figures:

- Minor edits to the text for grammar and clarity.
- Minor edits to figures (Fig. 1 made slightly larger, uniform font sizes in all figures).
- Added a reference to Caesar et al. (2018), on recent AMOC slowdown.
- Author Jo Brendryen moved from 5th to 3rd spot on the author list, due to his major work on the chronology section.

Finally, we wish to thank the reviewers and the editor for their work, and for considering our manuscript for publication in Nature Communications.

Respectfully,

Helsinki, 27 April 2018

Dr. J. Sakari Salonen, on behalf of the co-authors

E-mail: sakari.salonen@helsinki.fi

References

- Caesar L, Rahmstorf S, Robinson A, Feulner G, Saba V (2018) Observed fingerprint of a weakening Atlantic Ocean overturning circulation. *Nature* 556:191–196.
- Dutton A, Lambeck K (2012) Ice Volume and Sea Level During the Last Interglacial. *Science* 337:216–219.
- Eronen M, Ristaniemi O (1992) Late Quaternary crustal deformation and coastal changes in Finland. *Quaternary International* 15/16:175–184.
- Helmens K, *et al.* (2015) Major cooling intersecting peak Eemian Interglacial warmth in Northern Europe. *Quaternary Science Reviews* 122:293–299.
- Hoffman JS, Clark PU, Parnell AC, He F (2017) Regional and global sea-surface temperatures during the last interglaciation. *Science* 355:276–279.
- Juggins S (2013) Quantitative reconstructions in palaeolimnology: New paradigm or sick science? *Quaternary Science Reviews* 64:20–32.
- Kopp RE, Simons FJ, Mitrovica JX, Maloof AC, Oppenheimer M (2009) Probabilistic assessment of sea level during the last interglacial stage. *Nature* 462:863–868.
- Mokkedem Z, McManus JF, Oppo DW (2014) Oceanographic dynamics and the end of the last interglacial in the subpolar North Atlantic. *PNAS* 111:11263–11268.
- Moseley GE, Spötl C, Cheng H, Boch R, Min A, Edwards RL (2015) Termination-II interstadial/stadial climate change recorded in two stalagmites from the north European Alps. *Quaternary Science Reviews* 127:229–239.
- Pliik A, *et al.* (2016) Development of an Eemian (MIS 5e) Interglacial paleolake at Sokli (N Finland) inferred using multiple proxies. *Palaeogeography, Palaeoclimatology, Palaeoecology* 464:11–26.

- Salonen JS, Seppä H, Luoto M, Bjune A, Birks HJB (2012) A North European pollen–climate calibration set: analysing the climate response of a biological proxy using novel regression tree methods. *Quaternary Science Reviews* 45:95–110.
- Salonen JS, Luoto M, Alenius T, Heikkilä M, Seppä H, Telford RJ, Birks HJB (2014) Reconstructing Late-Quaternary climatic parameters of northern Europe from fossil pollen using boosted regression trees: comparison and synthesis with other quantitative reconstruction methods. *Quaternary Science Reviews* 88:69–81.
- Sykes MT, Prentice IC, Cramer W (1996) A bioclimatic model for the potential distributions of north European tree species under present and future climates. *Journal of Biogeography* 3:203–233.
- Vansteenberghe S, Verheyden S, Cheng H, Edwards RL, Keppens E, Claeys P (2016) Paleoclimate in continental northwestern Europe during the Eemian and early Weichselian (125–97 ka): insights from a Belgian speleothem. *Climate of the Past* 12:1445–1458.
- Vansteenberghe S, *et al.* (2017) The last glacial inception in continental northwestern Europe: characterization and timing of the Late Eemian Aridity Pulse (LEAP) recorded in multiple Belgian speleothems. *Geophysical Research Abstracts* 19:EGU2017-980.
- Zhuravleva A, Bauch HA, Van Nieuwenhove N (2017) Last Interglacial (MIS5e) hydrographic shifts linked to meltwater discharges from the East Greenland margin. *Quaternary Science Reviews* 164:95–109.

Reviewers' comments:

Reviewer #1 (Remarks to the Author):

Only minor comment.

WAPLS 3 for TJan is likely an overfitted model, perhaps because of spatial autocorrelation in the calibration set. I may have missed it, how were the model parameters chosen. Does accounting for autocorrelation, for example with h-block crossvalidation change the model performance substantially?

Reviewer #2 (Remarks to the Author):

I have read with interest the revised manuscript by Salonen et al. Overall, I am satisfied with most of the answers provided to the issues I raised in my first review except for a few that I will reiterate below. I would strongly ask the authors to take them into consideration. I also list a few additional minor changes that I would like them to do.

- While I appreciate that the authors are now referring to the latest marine temperature synthesis for the LIG by Hoffman et al. (2017), they have to be very careful in the way they refer to this study and associated LIG surface temperature estimate. The estimate proposed in this study is a global SEA surface temperature estimate and thus, does not include terrestrial climate reconstructions. I think this is crucial that this is pointed out in the manuscript if the authors are to refer to the $+0.5\pm 0.3^{\circ}\text{C}$ number. In other words they cannot simply refer to it as a global estimate e.g. "...the Eemian was characterized by temperatures above preindustrial levels by an estimate 0.5°C globally". Also they need to provide the associated uncertainty to this estimate of $\pm 0.3^{\circ}\text{C}$. Finally it is also important to mention that this estimate is interpreted as an annual signal while the estimates for the Arctic by CAPE Last Interglacial Project Members (2006) are interpreted as a summer estimate. The sentence starting line 52 by "Based on geological proxy data..." thus needs to be reformulated.

Also I agree with the authors that we lack of direct geological evidence regarding the contribution of Antarctica to the LIG sea level rise and that the Dutton et al. (2015) refers to a "substantial (but not complete) retreat of the southern sector [of the GIS]" during the LIG. Still I would argue that considering the large range of Greenland contribution to LIG sea level (0.6 to 3.5 m stated in Dutton et al. 2015 and details are given in Figure 3 of this paper), the use of the word "significant" should not be used in the expression in Line 54 "...and significantly reduced polar ice sheets in Greenland and possibly western Antarctica".

In order to account for those two comments, I would ask the authors to reformulate their sentence starting line 52 as such (now two sentences are proposed).

"Marine sediment records suggest that the Eemian was characterized by an average global annual sea surface temperature of $+0.5\pm 0.3^{\circ}\text{C}$ above preindustrial levels (Hoffman et al. 2017) and by summer surface warming of up to $4\text{-}5^{\circ}\text{C}$ above Arctic lands (CAPE LIG project members 2006). Also, Eemian global sea level was 6 to 9m above present associated with reduced polar ice sheets (Dutton et al. 2015)."

- I reiterate my comment regarding the very awkward formulation "Eemian Interglacial period" proposed in the title, abstract and beginning of the paper. Please do not use this expression. The scientific community has introduced enough different expressions to refer to this time interval (Last Interglacial, last interglacial period, last interglaciation, Eemian...). I truly believe that we do not need a new expression such as the one proposed by the authors. So again, I would ask the authors not to use it. In the title they should either refer to "the Last Interglacial" or the "Eemian". In the abstract and the introduction they should introduce the time interval with a sentence along

those lines:

"The Last Interglacial (ca 129,000 to 116,000 years ago, hereafter referred to as the Eemian)"

- Additional changes:

- o Line 42: "North Atlantic OCEANIC circulation regime"

- o Line 61: "...climate warming RELATIVE TO PREINDUSTRIAL"

- o Line 64: "Despite decades of work on the Eemian, EXISTING CLIMATE SYNTHESSES CONTINUE to be..."

- o Line 77: "...and/or significantly delayed onset of interglacial conditions" is an unclear and a misleading statement. Please replace by "AND ASYNCHRONOUS HEMISPHERIC SURFACE TEMPERATURE CHANGES".

- o Line 108: Replace "most important" by "key".

- o Line 110: "seven" instead of "7".

- o Line 195: analogue without "s".

- o Line 255: δ

- o Line 257: "...indicate THE AGE AND ASSOCIATED 2σ UNCERTAINTY for the onset of ..."

- o Line 268: "summer TEMPERATURE reconstruction..."

- o Line 339: "has not included the impact of meltwater pulses or...". Loutre et al. (2014) did some transient simulations with freshwater forcing at the onset of the LIG. I appreciate this is different from testing abrupt changes within the LIG, but still, it should not be forgotten and thus, the statement should be reformulated.

- o Line 359: "warmer than PREINDUSTRIAL conditions".

- o Line 412: "EEMIAN CLIMATE"

References:

Dutton et al. Science. Sea-level rise due to polar ice-sheet mass loss during past warm periods. Science, 349(6244), 2015.

Hoffman et al. 2017. Regional and global sea-surface temperatures during the last interglaciation. Science 355, 276-279.

Loutre et al. 2014. Factors controlling the last interglacial climate as simulated by LOVECLIM1.3. Clim. Past 10, 1541-1565.

Reviewer #3 (Remarks to the Author):

I reviewed a previous version of the manuscript and find it much easier to read. The paper is well-structured, well-focused, the goals are stated and fulfilled, figures are improved. The study elaborates on crucial chronological issues, suggests new age constraints for the Eemian sequence at the Sokli site, discusses climate variability and a complex interplay between different climatic forcings, affecting the last interglacial climate across entire Europe. The obtained dataset is unique; the new results provide strong evidence for the conclusions, albeit these are not completely novel, which is stated in the manuscript. Overall, the study is very important for scientists in specific fields and could be interesting for researchers in other related disciplines. Therefore, I would recommend this paper for publishing in Nature Communications.

I have, however, some minor reservations:

1. I believe that the main findings of the study (long-term temperature trends and their forcing mechanisms) could be better described in the Introduction. In particular, L. 38 "due to the mixed influence of insolation and oceanic forcing" appears too general, if consider the actual work that has been done.

2. With regard to the reliability of the presented Eemian Tjan reconstruction, it would be interesting to get some insights into Holocene Tjan reconstructions (if available), e.g., was there a comparable coupling between winter temperatures, insolation and oceanic forcing? On the other

hand, statistical significant testing - I have to acknowledge, however, that use of statistics is outside the scope of my expertise - as well as independent proxy-data support the presented winter temperature reconstruction. I also admit that a Holocene-Eemian comparison could be a focus of another separate study, published elsewhere.

3. I am still not very convinced about the suggested linkage between the two-pronged structure of the Tunturi event at Sokli and changes in the NADW formation, recorded in sediment data from the Eirik Drift (Ls. 313-325). Given that the overflows from the Nordic Seas established only during mid-Eemian (Hodell et al., 2009), early-Eemian $\delta^{13}\text{C}$ data from the southern Labrador Sea could largely reflect regional changes (Deaney et al., 2017), possibly with implications for the mid-latitude North Atlantic, but not for northern Europe. However, some climatic/oceanic variability does exist across the cooling intervals in the Sokli/Eirik Drift records and climatic connections are possible.

Some comments:

L. 50: "ka" should be introduced. In general, I would still suggest changing 129,000 to 116,000 years ago in the introduction (L. 31) to 129 to 116 thousand years ago.

L. 69: delete "the scarcity of"

L. 128: move "following the Tunturi event" to L. 127 (before "the onset of Zine V) or delete it
Ls. 128-130: "with pollen values" proportions, abundances...? "(ca. 1%) only seen...". The meaning is unclear.

L. 141: Change "all" to "the majority of" or "used"

L. 179-180: give reference "for Late Holocene samples" or move Fig. 3C to the end of the sentence

L. 204: change "caused" to "biased"

L. 241: Black arrows?

L. 245: NADW should be introduced

L. 246: Change the reference. The data from the East Greenland margin is not from Irvani et al., 2016.

L. 255 and L. 257: change d to $\langle\delta\rangle$

Ls. 282-285: Rephrasing might be useful. E.g., "Although the Tjan reconstructions should be interpreted with caution, due to $\langle\dots\rangle$, here we regard..."

Ls. 326-328: Is the spread of Arctic waters relevant here? It may be confusing for a reader that, on the one hand, "Arctic water spread in the subpolar North Atlantic" during the late Eemian (but apparently not in the eastern Nordic Seas?), and, on the other hand, "the warm Nordic Seas" (which is also subpolar North Atlantic) could explain high Tjan at Sokli site (L. 393).

Ls. 329-330: The statement needs rephrasing.

L. 337: What is meant by isolated pollen and speleothem sequences?

Ls. 338-340: I suggest rephrasing, otherwise a logical transition/conjunction with the passage above is missing. E.g., In this venue it is important to note... that models do not reproduce short-term climatic shifts, as they do not include the impact of glacier meltwater...

L. 344: a northern distribution of the events

L. 400: Is referring to Bakker et al. (2014) here correct?

L. 482: "onset" could be used instead of "base"

Ls. 498-500: two sentences could be combined (e.g., by using "by analogy to the last deglaciation")

Ls. 517-518: The meaning of the sentence is not clear. The presented reconstructions, however, do show temperatures warmer than present for the terminal part of the Eemian.

Supplementary information

- Change the order of authors

References

Hodell, D.A., Minth, E.K., Curtis, J.H., McCave, I.N., Hall, I.R., Channell, J.E. and Xuan, C., 2009. Surface and deep-water hydrography on Gardar Drift (Iceland Basin) during the last interglacial

period. *Earth and Planetary Science Letters*, 288(1-2), pp.10-19.

Deaney, E.L., Barker, S. and Van De Flierdt, T., 2017. Timing and nature of AMOC recovery across Termination 2 and magnitude of deglacial CO₂ change. *Nature communications*, 8, p.14595.

Reviewer 1: *Only minor comment. WAPLS 3 for TJan is likely an overfitted model, perhaps because of spatial autocorrelation in the calibration set. I may have missed it, how were the model parameters chosen. Does accounting for autocorrelation, for example with h-block crossvalidation change the model performance substantially?*

The 3-component WA-PLS model for T_{jan} was selected based on the van der Voet (1994) t -test, as implemented in the `rand.t.test` function of the `rioja` R package.

The reviewer asks about h -block cross-validation, a highly robust CV approach where the calibration set excludes all samples within a specified geographic distance (h -block radius) of the test sample. We have tested these models in h -block CV. These results did not make it in the present paper, due to the large computing demands which required a supercomputer run.

[Redacted].

[Redacted]

We also tested the T_{jan} reconstruction using the 2- and 3-component WA-PLS models. The reconstructions are very similar (see figure, below). In the multi-model T_{jan} ensemble WA-PLS was an outlier, showing persistently higher temperatures compared to other reconstructions. In this regard the 2-component model shows somewhat worse agreement still, with slightly higher temperatures compared to the 3-component model.

While we understand the reviewer's concern over a 3-component WA-PLS model, it appears the choice between the 2- and 3-component models is not critical for this paper.

Reviewer 2: *I have read with interest the revised manuscript by Salonen et al. Overall, I am satisfied with most of the answers provided to the issues I raised in my first review except for a few that I will reiterate below. I would strongly ask the authors to take them into consideration. I also list a few additional minor changes that I would like them to do.*

- *While I appreciate that the authors are now referring to the latest marine temperature synthesis for the LIG by Hoffman et al. (2017), they have to be very careful in the way they refer to this study and associated LIG surface temperature estimate. The estimate proposed in this study is a global SEA surface temperature estimate and thus, does not include terrestrial climate reconstructions. I think this is crucial that this is pointed out in the manuscript if the authors are to refer to the $+0.5\pm 0.3^{\circ}\text{C}$ number. In other words they cannot simply refer to it as a global estimate e.g. “...the Eemian was characterized by temperatures above preindustrial levels by an estimate 0.5°C globally”. Also they need to provide the associated uncertainty to this estimate of $\pm 0.3^{\circ}\text{C}$. Finally it is also important to mention that this estimate is interpreted as an annual signal while the estimates for the Arctic by CAPE Last Interglacial Project Members (2006) are interpreted as a summer estimate. The sentence starting line 52 by “Based on geological proxy data...” thus needs to be reformulated.*

Also I agree with the authors that we lack of direct geological evidence regarding the contribution of Antarctica to the LIG sea level rise and that the Dutton et al. (2015) refers to a “substantial (but not complete) retreat of the southern sector [of the GIS]” during the LIG. Still I would argue that considering the large range of Greenland contribution to LIG sea level (0.6 to 3.5 m stated in Dutton et al. 2015 and details are given in Figure 3 of this paper), the use of the word “significant” should not be used in the expression in Line 54 “...and significantly reduced polar ice sheets in Greenland and possibly western Antarctica”.

In order to account for those two comments, I would ask the authors to reformulate their sentence starting line 52 as such (now two sentences are proposed).

“Marine sediment records suggest that the Eemian was characterized by an average global annual sea surface temperature of $+0.5\pm 0.3^{\circ}\text{C}$ above preindustrial levels (Hoffman et al. 2017) and by summer surface warming of up to $4\text{--}5^{\circ}\text{C}$ above Arctic lands (CAPE LIG project members 2006). Also, Eemian global sea level was 6 to 9m above present associated with reduced polar ice sheets (Dutton et al. 2015).”

We agree that the range of uncertainties allows for a relatively minor reduction of GIS (i.e., 0.6 m sea-level equivalent). We also agree it is good to note the respective data types (marine vs. terrestrial) and climate variables used.

We thank the reviewer for the proposed summary of the available proxy evidence. We have adopted this in our revised manuscript, with the minor of addition of noting that the evidence suggests “reduced ice sheets in Greenland and possibly in western Antarctica” (but as requested by the reviewer, not “significantly reduced”). This relays the uncontroversial information that some reduction took place in Greenland, which is relevant for our Discussion.

Reviewer 2: • *I reiterate my comment regarding the very awkward formulation “Eemian Interglacial period” proposed in the title, abstract and beginning of the paper. Please do not use this expression. The scientific community has introduced enough different expressions to refer to this time interval (Last Interglacial, last interglacial period, last interglaciation, Eemian...). I truly believe that we do not need a new expression such as the one proposed by the authors. So again, I would ask the authors not to use it. In the title they should either refer to “the Last Interglacial” or the “Eemian”. In the abstract and the introduction they should introduce the time interval with a sentence along those lines:*

“The Last Interglacial (ca 129,000 to 116,000 years ago, hereafter referred to as the Eemian)”

As requested, we now only use the terms “Eemian” and “Last Interglacial”. We primarily use the “Eemian”, but on first use in Abstract and Introduction also give the alternative name “Last Interglacial” in parentheses.

Reviewer 2: *Additional changes:*

Line 42: “North Atlantic OCEANIC circulation regime”

Line 61: “...climate warming RELATIVE TO PREINDUSTRIAL”

Line 64: “Despite decades of work on the Eemian, EXISTING CLIMATE SYNTHESSES CONTINUE to be...”

Line 77: “...and/or significantly delayed onset of interglacial conditions” is an unclear and a misleading statement. Please replace by “AND ASYNCHRONOUS HEMISPHERIC SURFACE TEMPERATURE CHANGES”.

Line 108: Replace “most important” by “key”.

Line 110: “seven” instead of “7”.

Line 195: analogue without “s”.

Line 255: δ

Line 257: “...indicate THE AGE AND ASSOCIATED 2σ UNCERTAINTY for the onset of ...”

Line 268: “summer TEMPERATURE reconstruction...”

Done (all of the above).

Reviewer 2: *Line 339: “has not included the impact of meltwater pulses or...”. Loutre et al. (2014) did some transient simulations with freshwater forcing at the onset of the LIG. I appreciate this is different from testing abrupt changes within the LIG, but still, it should not be forgotten and thus, the statement should be reformulated.*

The reviewer is correct that Loutre et al. (2014) include early-Eemian freshwater fluxes (as does one model included in Bakker et al. (2013)). We have rephrased this statement to clarify that we mean “meltwater pulses occurring after the major early-Eemian deglaciation”. We have added a reference to Loutre et al. (2014) here.

Reviewer 2: *Line 359: “warmer than PREINDUSTRIAL conditions”.*
Line 412: “EEMIAN CLIMATE”

Done.

Reviewer 3: *I reviewed a previous version of the manuscript and find it much easier to read. The paper is well-structured, well-focused, the goals are stated and fulfilled, figures are improved. The study elaborates on crucial chronological issues, suggests new age constraints for the Eemian sequence at the Sokli site, discusses climate variability and a complex interplay between different climatic forcings, affecting the last interglacial climate across entire Europe. The obtained dataset is unique; the new results provide strong evidence for the conclusions, albeit these are not completely novel, which is stated in the manuscript. Overall, the study is very important for scientists in specific fields and could be interesting for researchers in other related disciplines. Therefore, I would recommend this paper for publishing in Nature Communications.*

I have, however, some minor reservations:

1. I believe that the main findings of the study (long-term temperature trends and their forcing mechanisms) could be better described in the Introduction. In particular, L. 38 "due to the mixed influence of insolation and oceanic forcing" appears too general, if consider the actual work that has been done.

We have modified this passage to be more specific and by eliminating unnecessary words. Especially, we clearly state that the summer and winter trends are opposite, with the former falling and the latter rising. The signs of the first-order seasonal trends are commonly assessed in palaeoclimate modelling, and are worth including here, in the Abstract. We are constrained by the maximum length of the Abstract in expanding this part.

Reviewer 3: *2. With regard to the reliability of the presented Eemian Tjan reconstruction, it would be interesting to get some insights into Holocene Tjan reconstructions (if available), e.g., was there a comparable coupling between winter temperatures, insolation and oceanic forcing? On the other hand, statistical significant testing - I have to acknowledge, however, that use of statistics is outside the scope of my expertise - as well as independent proxy-data support the presented winter temperature reconstruction. I also admit that a Holocene-Eemian comparison could be a focus of another separate study, published elsewhere.*

In this part of the world (~northeast European Arctic), the Holocene is very different: most reconstructions and transient modellings of summer and winter temperature show similar patterns between the seasons (e.g. Davis et al. 2003, Bakker et al. 2014). In the Arctic specifically, the modelling generally shows Arctic winter temperatures following summer temperature, due to sea-ice formation providing a “memory” of summer conditions during the following winter (Bakker et al. 2014).

The distinct Eemian decoupling might be due to a specific set of circumstances. A fall in T_{Jul} is forced by a massive fall in summer insolation over the Eemian. Meanwhile, an opposite T_{Jan} trend is forced not only by a rise in winter insolation, but also by a persistent vigorous AMOC into the late Eemian, together sufficient to produce a rising T_{Jan} trend during the insolation-driven minimum in T_{Jul} . We agree, however, that such comparisons between interglacials should be reserved to another paper.

Reviewer 3: *3. I am still not very convinced about the suggested linkage between the two-pronged structure of the Tunturi event at Sokli and changes in the NADW formation, recorded in sediment data from the Eirik Drift (Ls. 313-325). Given that the overflows from the Nordic Seas established only during mid-Eemian (Hodell et al., 2009), early-Eemian $\delta^{13}\text{C}$ data from the southern Labrador Sea could largely reflect regional changes (Deaney et al., 2017), possibly with implications for the mid-latitude North Atlantic, but not for northern Europe. However, some climatic/oceanic variability does exist across the cooling intervals in the Sokli/Eirik Drift records and climatic connections are possible.*

We agree the causal link between the two-pronged local event in the Labrador Sea, and the similarly structured Tunturi event, is quite speculative.

We have removed the closing statement of this passage, from L323–325, suggesting a causal link between the specific GIS/LIS outbursts and the two possible peaks of the Tunturi event. We now simply note the less controversial features – that there is a similar-duration cooling event between the Early and Late Eemian warm periods both in Sokli and in the Labrador and Nordic Seas, and note positions of the identified the meltwater outbursts.

Reviewer 3: *Some comments:*

L. 50: "ka" should be introduced. In general, I would still suggest changing 129,000 to 116,000 years ago in the introduction (L. 31) to 129 to 116 thousand years ago.

Done; we now use “thousand years” in Abstract and “thousand years (ka)” in Introduction.

Reviewer 3: *L. 69: delete "the scarcity of"*

Done.

Reviewer 3: *L. 128: move "following the Tunturi event" to L. 127 (before "the onset of Zine V) or delete it*

Deleted as suggested.

Reviewer 3: *Ls. 128-130: "with pollen values" proportions, abundances...? "(ca. 1%) only seen...". The meaning is unclear.*

Rephrased as “relative pollen abundances”.

Reviewer 3: *L. 141: Change "all" to "the majority of" or "used"*

Rephrased as “the examined proxies”.

Reviewer 3: L. 179-180: give reference "for Late Holocene samples" or move Fig. 3C to the end of the sentence

We have moved the figure reference to the end of the sentence.

Reviewer 3: L. 204: change "caused" to "biased"

Done.

Reviewer 3: L. 241: Black arrows?

Fixed.

Reviewer 3: L. 245: NADW should be introduced

Done; "North Atlantic Deep Water" spelled out.

Reviewer 3: L. 246: Change the reference. The data from the East Greenland margin is not from Irvani et al., 2016.

Fixed reference to Zhuravleva et al. (2017) – we thank the reviewer for spotting this one.

Reviewer 3: L. 255 and L. 257: change d to

Fixed d → δ .

Reviewer 3: Ls. 282-285: Rephrasing might be useful. E.g., "Although the Tjan reconstructions should be interpreted with caution, due to <...>, here we regard..."

Done.

Reviewer 3: Ls. 326-328: Is the spread of Arctic waters relevant here? It may be confusing for a reader that, on the one hand, "Arctic water spread in the subpolar North Atlantic" during the late Eemian (but apparently not in the eastern Nordic Seas?), and, on the other hand, "the warm Nordic Seas" (which is also subpolar North Atlantic) could explain high Tjan at Sokli site (L. 393).

This reference (L326–328) was included here as one of the studies showing events of prominent and increasing oceanic instabilities towards the Late Eemian. The authors cited here (Mokkedem et al. 2014) hypothesize a causal link, involving the Subpolar and Subtropical Gyres, between Arctic and Polar Front advances and the Late-Eemian intensification of the northern AMOC. However, we feel going into these oceanic circulation mechanisms would be beyond the scope of a terrestrial palaeoclimate paper such as this. The latter reference (L393) only considers the Nordic Seas temperatures, which is the proximally relevant marine variable in terms of Fennoscandian climate.

We have modified the wording in the manuscript, to emphasize that the frontal movements occur as transient events. We also see events in the Sokli record that may correspond with these events, and thus there is no contradiction.

Reviewer 3: *Ls. 329-330: The statement needs rephrasing.*

Rephrased to be more straightforward.

Reviewer 3: *L. 337: What is meant by isolated pollen and speleothem sequences?*

We use “isolated” here with roughly the same meaning as “sporadic” or “occasional”. Abrupt events are seen in some mid-latitude European records – cited here – but they are not seen in a large majority of records.

Reviewer 3: *Ls. 338-340: I suggest rephrasing, otherwise a logical transition/conjunction with the passage above is missing. E.g., In this venue it is important to note... that models do not reproduce short-term climatic shifts, as they do not include the impact of glacier meltwater...
L. 344: a northern distribution of the events*

Done.

Reviewer 3: *L. 400: Is referring to Bakker et al. (2014) here correct?*

This reference should have been to Risebrobakken et al. (2007); fixed.

Reviewer 3: *L. 482: "onset" could be used instead of "base"
Ls. 498-500: two sentences could be combined (e.g., by using "by analogy to the last deglaciation")*

Done.

Reviewer 3: *Ls. 517-518: The meaning of the sentence is not clear. The presented reconstructions, however, do show temperatures warmer than present for the terminal part of the Eemian.*

This statement refers to the Brendryen et al. (2010) study from eastern Norwegian Sea, proximal to the Fennoscandian land mass, cited a few lines earlier. We have corrected the sentence to specify “eastern Norwegian Sea” instead of “Nordic Seas”, and repeat the reference to Brendryen et al. (2010).

Reviewer 3: *Supplementary information
- Change the order of authors*

Done.

Some further minor edits have been done:

- Corrections to affiliations

Finally, we wish to thank all the reviewers for their detailed and thoughtful comments which we think have greatly improved the manuscript. We also thank the editor for his work.

Respectfully,

Helsinki, 7 June 2018

Dr. J. Sakari Salonen, on behalf of the co-authors

E-mail: sakari.salonen@helsinki.fi

References

- Bakker P, *et al.* (2013) Last interglacial temperature evolution – a model inter-comparison. *Climate of the Past* 9:605–619.
- Bakker P, *et al.* (2014) Temperature trends during the Present and Last Interglacial periods – a multi-model-data comparison. *Quaternary Science Reviews* 99:224–243.
- Brendryen J, Hafliðason H, Sejrup HP (2010) Norwegian Sea tephrostratigraphy of marine isotope stages 4 and 5: Prospects and problems for tephrochronology in the North Atlantic region. *Quaternary Science Reviews* 29:847–864.
- Davis BAS, Brewer S, Stevenson AC, Guiot J, Data Contributors (2003) The temperature of Europe during the Holocene reconstructed from pollen data. *Quaternary Science Reviews* 22:1701–1716.
- Loutré MF, Fichet T, Goosse H, Huybrechts P, Goelzer H, Capron E (2014) Factors controlling the last interglacial climate as simulated by LOVECLIM1.3. *Climate of the Past* 10:1541–1565.
- Mokkedem Z, McManus JF, Oppo DW (2014) Oceanographic dynamics at the end of the last interglacial in the subpolar North Atlantic. *PNAS* 111:11263–11268.
- Risebrobakken B, Dokken T, Otterå OH, Jansen E, Gao Y, Drange H (2007) Inception of the Northern European ice sheet due to contrasting ocean and insolation forcing. *Quaternary Research* 67:128–135.
- van der Voet H (1994) Comparing the predictive accuracy of models using a simple randomization test. *Chemometrics and Intelligent Laboratory Systems* 25:313–323.
- Zhuravleva A, Bauch HA, Van Nieuwenhove N (2017) Last Interglacial (MIS5e) hydrographic shifts linked to meltwater discharges from the East Greenland margin. *Quaternary Science Reviews* 164:95–109.